# Study on deformation of tunnel pile foundation based on discrete element method and finite difference method

Zhenchu Zhao[1], Yuan Zhang[2], Fang Dai[1]*

1 School of Civil Engineering, Central South University, Changsha, China, 2 Power China, Zhongnan Engineering Co., Ltd., Changsha, China

* dailyn2000@csu.edu.cn

## Abstract

The deformation of pile caused by tunnel excavation will weaken the bearing capacity of the foundation. In order to investigate the deformation response of pile induced by the construction of three-hole small spacing tunnel, the DEM-FDM (discrete element method and finite difference method) coupling numerical simulation method were used to simulate the deformation process of pile during tunnel excavation. This paper probed into the deformation response of pile by three factors: the length of pile, the pile-tunnel spacing, and the three-hole tunnel construction. The results showed that, as the pile-tunnel spacing decreases, the incremental horizontal displacement of the pile top became more significant when the three-hole tunnel was excavated. The excavation resulting in four zones of horizontal displacement concentration. The prominent locations were mainly concentrated on both sides of the tunnel and the ground directly above the tunnel. The research findings of this study can provide insights and references for the design and construction of shield tunneling under passing piles.

## Introduction

With the improvement of subway network in large cities, an increasing number of dense subway tunnels are passing through many building foundations. Currently, a large number of research works [1–3] and engineering practices has demonstrated that tunnel construction inevitably exerts a significant impact on overlying strata, resulting in surface settlement. Currently, researchers have studied the deformation of pile deformation and surface settlement induced by tunnel construction through various methods, including theoretical analysis [4–6], numerical simulation [3, 7, 8] and machine learning [9, 10].

Numerical simulation method with its unique advantages, can be well combined with the actual engineering situation, and the actual results of the project with a high degree of consistency, in order to become a more mature mainstream method. Among them, finite difference method (FDM) and finite element method (FEM) as continuous medium method are mainly utilized in numerical simulation. Huang et al. [11] established a 3D FEM model to analyze the

**Data Availability Statement:** All relevant data are within the manuscript and its Supporting Information files.

**Funding:** This paper was supported by some funds from the National Natural Science Foundation of

China. The funds are numbered 51778633 and 51308552. Funders were involved in research data collection and analysis.

**Competing interests:** The authors have declared that no competing interests exist.

displacement of pile induced by double-tunnel construction under different construction sequences. The results show that the change of construction sequence of double-parallel tunnel has great influence on adjacent piles. That is, the settlement of piers can be reduced by 13.1% when the tunnel closer to the pile is built first, and then as the tunnel away from the pile is built. Marshall et al. [12] established the deformation effect of a new tunnel on adjacent pile by considering the influence of tunnel location on tunnel-pile interaction. Li et al. [8] used FLAC3D to conduct detailed numerical simulations of the construction process of pile and shield tunnel. The deformation of bridge pile is mainly caused by ground loss and excavation disturbance during shield tunneling. The settlement caused by pile support accounts for about 3%-20% of the total settlement. The reduction effects of settlement deformation, lateral displacement and principal stress mainly show on the supporting pile, but the change of non-supporting pile is the least. Through this research, the existing researches on the response induced by tunnel construction mainly focuses on the influence of surface settlement. The study of adjacent pile was also mainly aimed at the response induced by the construction of single tunnel or double tunnel [13–17]. However, the action mechanism and deformation response of the three-hole small spacing tunnel construction to the pile of adjacent buildings are still unclear.

Machine learning and numerical simulation are two commonly used methods [18–25]. Discrete element method (DEM), as an emerging numerical simulation method, have been applied in various fields. In geotechnical engineering, it. can simulate the interaction between particles or particle groups, and is suitable for simulating the behavior of granular materials such as soil. FDM, as a continuous medium method, can better describe the changes in continuous media and is more suitable for concrete continuity modeling. By coupling two methods, the interaction between granularity and continuity can be considered comprehensively, so that the behavior of pile in the process of excavation can be simulated more accurately. The DEM-FDM coupling method was verified by Gholaminejad et al [26]. It is a robust method to simulate geotechnical problems. lee et al. [27] used DEM and FDM to conduct 3D numerical simulation of earth pressure balance (EPB) shield tunnel. The simulation results show that the numerical model based on DEM-FDM coupling can reasonably simulate tunnel shield tunneling under the condition of TBM operation, and has good robustness. Consequently, more and more research has based on DEM-FDM coupling method [28–30]. Qu et al. [31] established a computational model using the coupled DEM-FDM method and realized the simulation of the whole process of shield construction. By using the coupling method, better results can be obtained in the study of pile and tunnel [32, 33].

In order to further explore the deformation response mechanism of pile induced by the construction of a three-hole small distance tunnel, this paper conducted further parametric modeling calculations using DEM-FDM coupling method, established numerical simulation models and conduct parametric analysis. The analysis focused on the deformation and stress responses of adjacent piles with varying lengths and distances, sequentially affected by the excavation of the three-hole small distance tunnel.

## Research method

### DEM-FDM coupling model and parameter setting

In order to investigate the soil deformation caused by subway tunnel excavation and the response of pile foundation deformation, this paper selects the finite difference program FLAC2D to build tunnel segments and adjacent pile foundation through grid modeling. PFC2D is used to deposit strata through particles. As the problems in computational efficiency of 3D computational analysis have not been solved, this paper currently discusses the

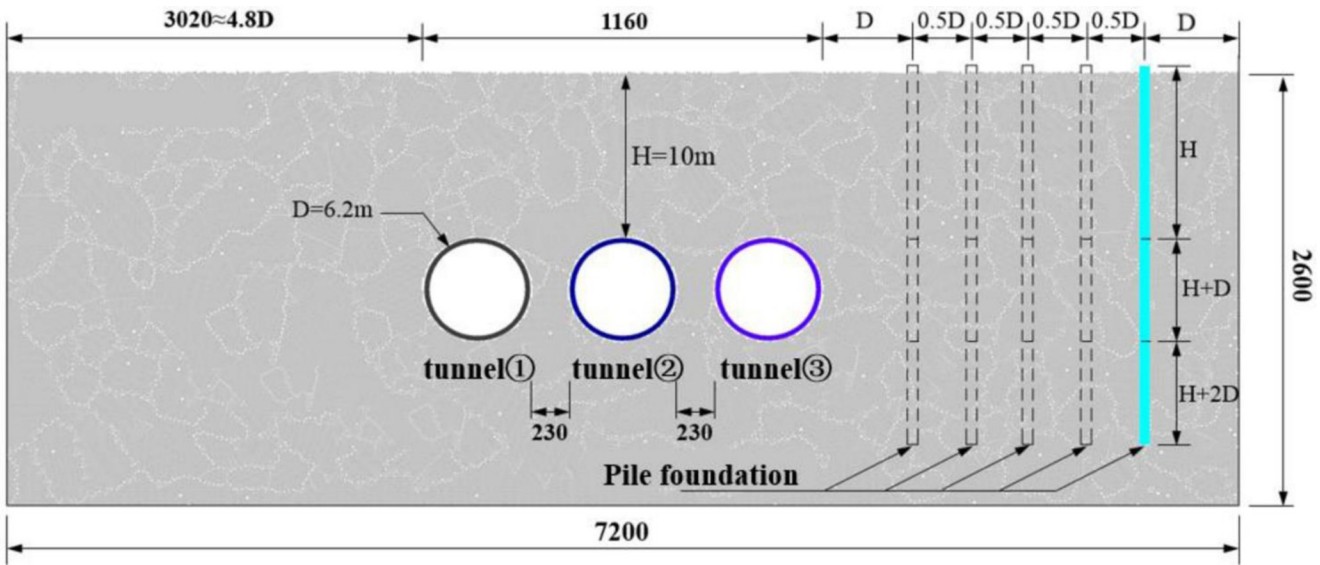

**Fig 1. DEM—FDM coupling calculation model of shield tunnel.**

horizontal deformation of piles only at the 2D level. A two-dimensional computational stratigraphic model with a width of 72m and a height of 26m was established, as shown in Fig 1. The diameter of the tunnel excavation is 6.4m, the outer diameter of the segment is 6.2m, that is, the shield tail clearance is 10cm, the soil loss rate is 6.15%, and the thickness of the segment is 35cm according to the actual construction situation. In the process of accumulation modeling, in order to ensure the accuracy of particle part calculation. The number of particles should be reduced as much as possible to ensure that the computer can give full play to the calculation power. The particle size is 5cm, and the total number of particles before excavation is 84,117. In order to reflect the influence mechanism of buried depth of three-wire tunnel with small distance on pile foundation and stratum, the tunnel buried depth is set as 13.1m, the covering layer thickness at the top of the tunnel is 10m, the distance between the bottom of the segment and the bottom of the model is 10m (1.6D), and the distance between the outer side of the tunnel segments on both sides of the model is 25m (4D). Where, D was the tunnel diameter, 6.2m. It can ensure that the effect of model boundary does not affect the calculation result [34]. This chapter only discusses the geometric characteristics of pile foundation and the influence of the distance between pile foundation and tunnel, so the difference of different soil layers is ignored. The horizontal displacements of the piles obtained from the calculations may be large compared to the actual conditions because the effect of infill grouting on the soil loss rate was not considered. However, this does not affect the analysis of the effect of pile length and pile-tunnel spacing since the fill grouting was not considered for all the conditions.

The numerical simulation process in this paper satisfies the following assumptions:

1. Assuming that the surrounding soil is isotropic and homogeneous, this paper only discusses the geometric characteristics of pile foundation and the influence of the distance between pile foundation and tunnel, so the differences between different soil layers are ignored.

2. It is assumed that the influence of grouting on pile foundation deformation is not considered.

3. Assume that the construction sequence of the three tunnels is from one side to the other.

**Table 1. Soil layer parameter settings.**

| Soil property | $\gamma(kN/m^3)$ | Emod(Pa) | krat | pb_ten(N) | pb_coh(kPa) | Fric(°) |
|---|---|---|---|---|---|---|
| Sandy silt with silty sand | 18.5 | 4e7 | 1.5 | 7e6 | 9.4 | 27.8 |

In order to ensure that the model is fully balanced and particles complete stacking contact before calculation, soil particles are established from bottom to top layer before excavation calculation, and stacked layer by layer to balance. After the balance of particle accumulation is completed, the excess particles are deleted and the balance is continued to stabilize. During the balancing process, the particle contact force distribution and particle velocity were monitored to determine whether the model reached equilibrium. When the Ratio aver was less than 1e-5 and the particle velocity was less than 0.001 m/s, the model was considered to have been balanced.

In the DEM field, the contact parameters of particles were calibrated through a series of large-scale triaxial test analyses. In the process of numerical simulation in this paper, the selection of parameters was subject to field monitoring, and the calibration test was carried out by triaxial test, and the modeling parameters of the following soils were obtained. Triaxial test was a common method to determine the macroscopic parameters of soil. Unconsolidated and undrained tests were carried out on a series of sand and clay with density gradient through the triaxial tester. The relationship between the internal friction Angle $\phi$, cohesion force c, deformation modulus k, and porosity was fitted, and the correlation between the shear strength and porosity of the soil was studied, which provided a basis for the parameter assignment of the subsequent shield model.

The values of soil layer related parameters in the simulation calculation are shown in Table 1 [32], and the values of model structure parameters are shown in Table 2. Values in the table refer to engineering geotechnical test results. The linear model used in the Ball-facet is as follows: the normal stiffness kn is 1e7, and the tangential stiffness ks is 3e6. The linear bond contact parameters between Ball and ball are inherited by property and are consistent with the parameters of the particle itself, $k_n$ and $\bar{k}_n$ are 6.42e6, $k_s$ and $\bar{k}_s$ are 2.36e6, μ is 1.6, and damp is 0.6.

The lining segment is made of C50 concrete, the pile foundation is made of concrete structure, and the cap roof and pile foundation are made of C30 concrete. Select the calculation parameters according to the actual situation. Table 2 lists the calculation parameters [35].

The discrete element method in this paper also has three basic assumptions in the numerical simulation calculation process, which are as follows:

1. The particles in the discrete element model are all rigid bodies, and the deformation of the particle system is the sum of the deformation of the contact points between the particles;

2. Due to the small particle size, the contact range between particles is very small

3. The contact between particle units is flexible, and there is a certain overlap between particles, but relative to the size of the particles, the overlap range is very low.

**Table 2. Model structure parameter settings.**

| Structure name | Materials | Constitutive model | Bulk modulus (Pa) | Shear modulus (Pa) | $\rho(kg/m^3)$ |
|---|---|---|---|---|---|
| Pile foundation | C30 | Elastic model | 2.0e10 | 1.2 e10 | 2500 |
| lining | C50 | Elastic model | 1.917e10 | 1.438e10 | 2500 |

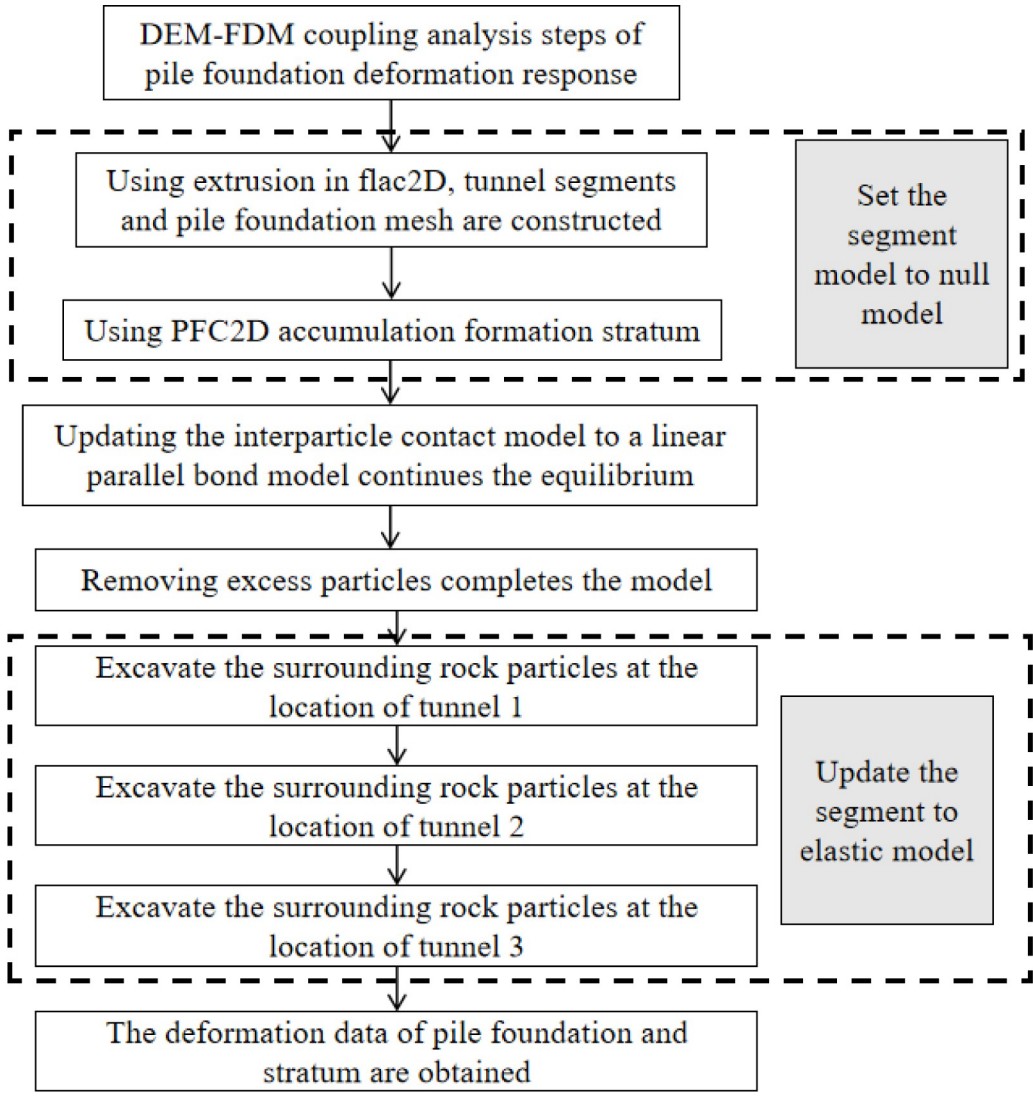

**Fig 2. Process of DEM-FDM modeling analysis.**

In order to ensure full settlement and compact arrangement of particles, tangential stiffness and critical damping are set to 0 and only normal stiffness is set. After particles are packed up, tangential stiffness and critical damping are assigned to continue the balance calculation. The steps of coupling calculation analysis are shown in Fig 2.

Firstly, the tunnel segment and pile foundation network are built by using extrusion in FLAC2D. At the same time, the segment constitutive model is set to null model. The second step is to use PFC2D deposit to generate strata. After that, the displacement constraint of pile foundation network is reduced, and the contact model between particles is updated to a linear parallel bond model. To achieve balance. The surrounding rock particles at the location of each tunnel are excavated successively, and the segments are updated into elastic models. Finally, the results of pile foundation and stratum deformation are derived.

## Model boundary setting and numerical simulation analysis scheme

For the DEM-FDM coupling simulation, the particle and mesh components can exchange forces and displacements (stress-strain) between each other. Therefore, at the contact interface

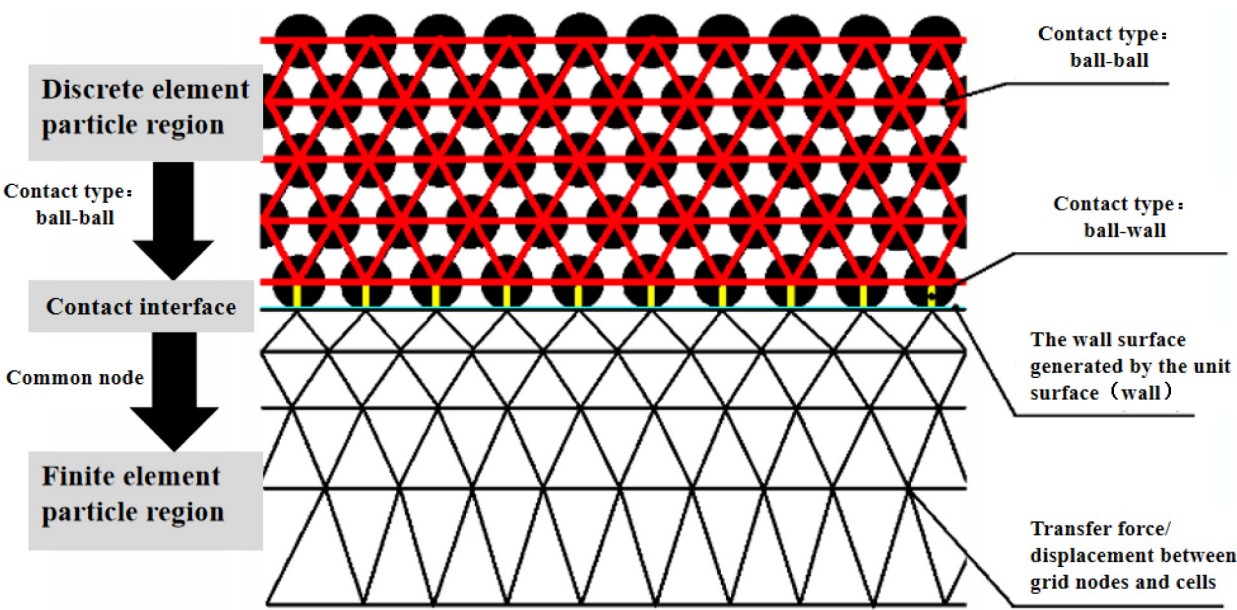

**Fig 3. Schematic diagram of DEM-FDM coupling model.**

between the cell grid and particles, they act as mutual boundaries. In the model established in this paper, there is no need for additional displacement constraints in the formal calculation of the pile foundation and segments. The soil layer within the granular part acts as an elastic boundary for the pile foundation and segments. Additionally, during the modeling and calculation process, the boundary constraint settings in the model are adjusted corresponding to different stages of the analysis. Some model units utilize FLAC2D's built-in Extrusion modeling module. Before the formal calculation, both the segment and pile foundation are modeled. To maintain soil stress balance, a Null constitutive model is initially set for the segment to mitigate the effects of lining on particle accumulation. Simultaneously, the pile foundation is fully constrained to simulate its state when buried in soil. Once the balance calculation is completed, the pile foundation constraints are entirely removed. The particles within the tunnel section are eliminated, and the segment's constitutive model is updated to an elastic constitutive model. Consequently, the surrounding rock of the particle section provides boundary conditions for the segment. The entire model is enclosed within a box generated by the wall, as the box's boundary acts as the grain boundary for the soil layer component. The DEM-FDM coupling model is shown in Fig 3.

The surface settlement caused by tunnel construction is mainly affected by tunnel size, soil loss rate, tunnel buried depth and soil layer properties. The additional effect of tunnel excavation on adjacent pile foundation is mainly affected by the relative position relationship between pile foundation and tunnel. Therefore, three pile lengths (pile length L equals buried depth H) are selected in this paper to analyze the situation where the tunnel is located below the pile end. Pile length L = H+D, used to analyze the tunnel at the depth of the pile end. And L = H+2D, which is used to analyze the situation when the tunnel is located on the side of the pile. The distance between tunnel and pile foundation is selected in the D-3D range. When the buried depth of the three-hole small distance tunnel is less than 10m, the width of the surface settlement trough caused by construction is about 8D. Considering that the distribution range of the surface settlement based on the project in this paper is about 3D on the outer edge of the tunnel on both sides, the setting calculation scheme is shown in Table 3.

**Table 3. Numerical simulation scheme of three-hole tunnel excavation with small distance.**

| Case | The length of pile foundation | Distance between tunnel and pile foundation |
|------|-------------------------------|---------------------------------------------|
| 1 | 10m (H) | 6.2m (D) |
| 2 | 16.2m (H+D) | 6.2m (D) |
| 3 | 22.4m (H+2D) | 6.2m (D) |
| 4 | 10m (H) | 9.3m (1.5D) |
| 5 | 16.2m (H+D) | 9.3m (1.5D) |
| 6 | 22.4m (H+2D) | 9.3m (1.5D) |
| 7 | 10m (H) | 12.4m (2D) |
| 8 | 16.2m (H+D) | 12.4m (2D) |
| 9 | 22.4m (H+2D) | 12.4m (2D) |
| 10 | 10m (H) | 15.5m (2.5D) |
| 11 | 16.2m (H+D) | 15.5m (2.5D) |
| 12 | 22.4m (H+2D) | 15.5m (2.5D) |
| 13 | 10m (H) | 18.6m (3D) |
| 14 | 16.2m (H+D) | 18.6m (3D) |
| 15 | 22.4m (H+2D) | 18.6m (3D) |

## Construction plan design

Take an urban rail transit shield section in China as an example, the shield is a three-hole section, including the up line, the down line and the stop line. The designed start and end points for the uprunning line in this section are: SK20+226.052 to SK20+741.314, spanning a total length of 515.262 m. Similarly, the designed start and end points for the downward line are: XK20+227.065 to XK20+741.330, covering a total length of 514.197 m. The start and end points for the parking line in this section are: TK0+112.500 to TK0+627.514, with a total length of 515.014 m. The vertical distance between the top and bottom lines is 22 m. Therefore, this tunnel is a small clear distance tunnel. The maximum and minimum slope of the line are 4‰ and 2‰, respectively. The burial depth of the line ranges from 8.916 to 9.946 m, and the outer diameter of the segment is 6.2 m.

The line of the three-hole shield tunnel section travels along the existing road. Along the line are the city's main roads and important commercial blocks and residential areas. The underground pipeline along the line is dense and passes through the existing civil air defense tunnel. The actual scene diagram of the project case is shown in Fig 4(A).

The soil quality of the shield section is mainly silt and sand. Therefore, the earth pressure balance shield machine is selected for construction. The shield machine can be applied to water-rich pebbles, silty sand stratum, soft and hard composite stratum, breezy rock stratum and silty clay. The diameter of the shield tunneling machine is 6480mm, and the panel box spoke cutter head is adopted. The length, inner diameter and outer diameter of the segment are 1200mm, 5500mm and 6200mm, respectively.

The construction sequence of shield tunnel sections is shown in Fig 4(B). The shield tunneling machine first excavates tunnel 1, starting from station B and finishing at station A. The parking line, tunnel 2, is then excavated, again starting from station B and finishing at station A. Finally, tunnel 3 is excavated, and the shield machine exits after receiving it from Station A.

Through the above engineering case data, the establishment of the numerical model was completed. The results of surface settlement monitoring and numerical simulation were compared to demonstrate the rationality of the model. This paper collected the actual data of the tunnel construction site, such as ground settlement change, pile foundation deformation, etc., and compared these data with the simulation results. The following is a comparison between the numerical simulation results of land surface settlement change and the field monitoring

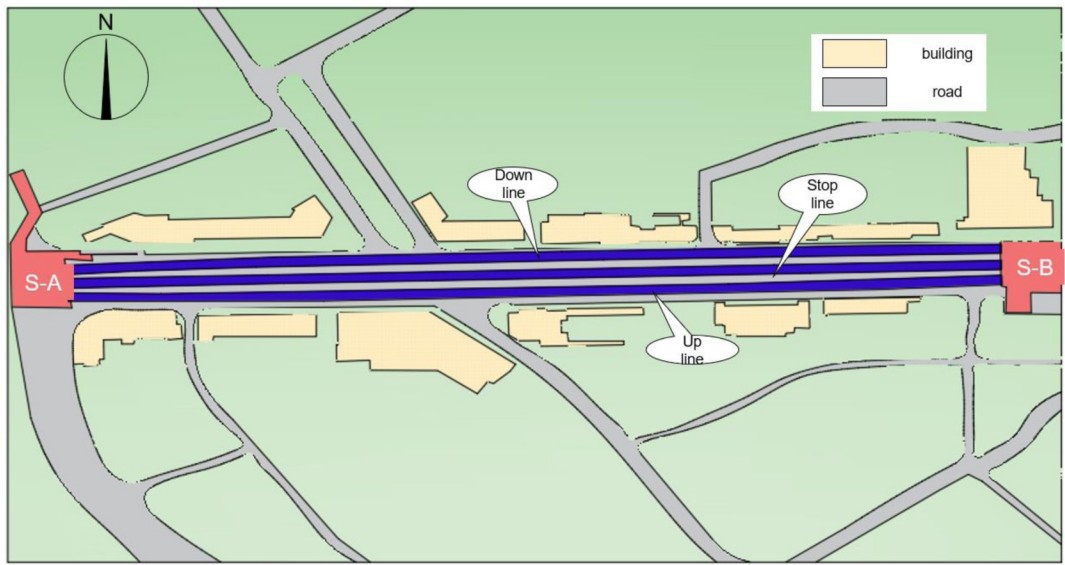

**(a) Project case site layout**

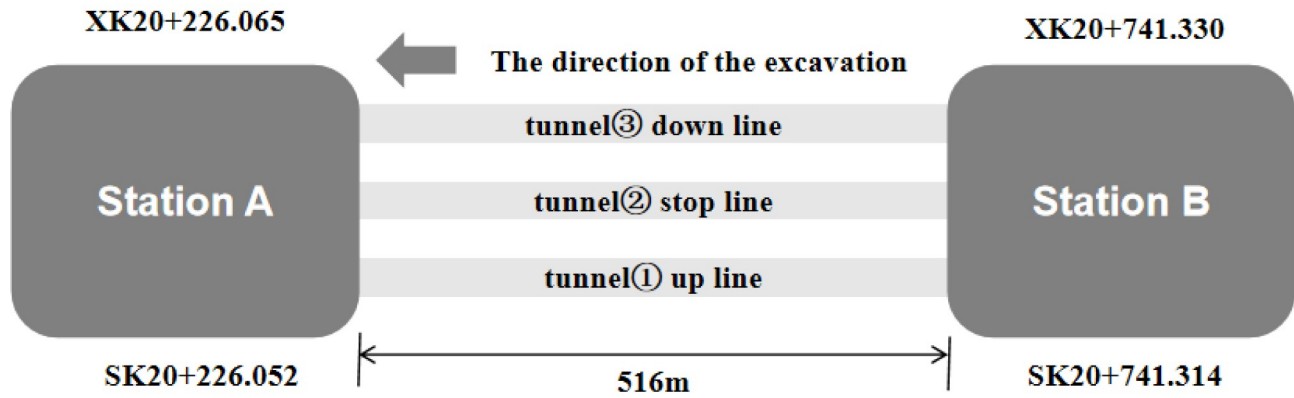

**(b) Construction sequence of shield tunnel sections**

Fig 4. **Engineering design scheme.**

results. The maximum settlement value obtained by numerical simulation was -11.8mm, and the maximum settlement value obtained by fitting with the measured fitting data was -11.63mm, with a comparison error of only 0.26%. As could be seen from the Fig 5 below, the numerical model presented in this paper had certain reliability, and the verification and comparison of surface settlement values showed that the model had a strong fitting ability.

## Results and discussion

### Effect of pile length

The length of pile is closely related to its bearing capacity and stability. This section analyzes the response rules of piles of different lengths to the construction of three-hole small distance tunnels.

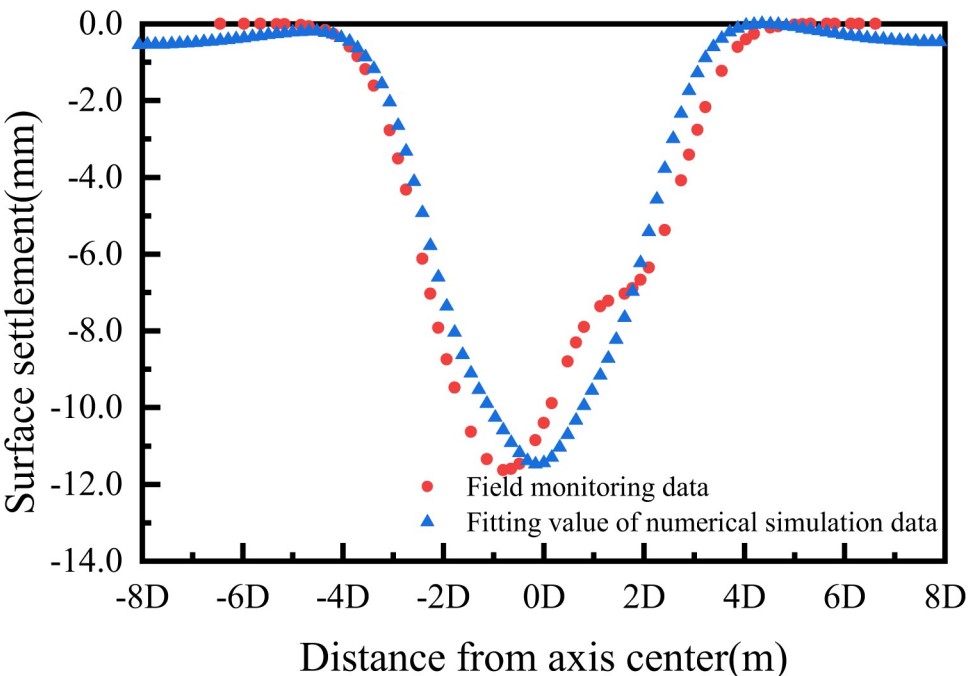

**Fig 5. The results of numerical simulation verification.**

Fig 6 shows the vertical displacement of the pile top and bottom under different conditions. The results indicate that tunnel construction can cause settlement or slight uplift of the piles. When the pile-tunnel distance is 1D, the pile with a length of H+D experiences the maximum settlement, which is only 3mm. Thus, the vertical displacement of the pile caused by tunnel

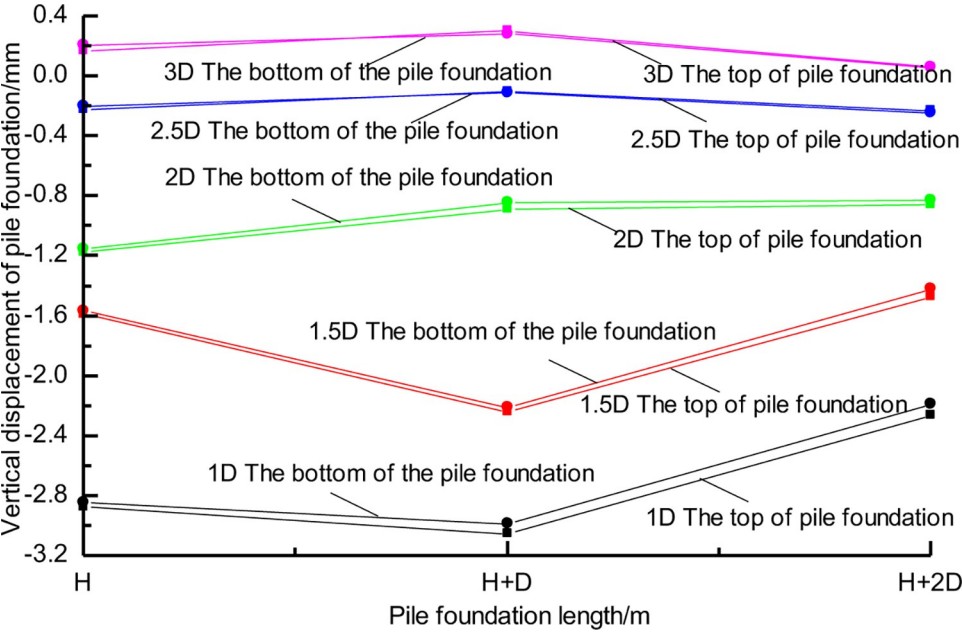

**Fig 6. Vertical displacement of pile foundation with different distance between three kinds of pile foundation and tunnel.**

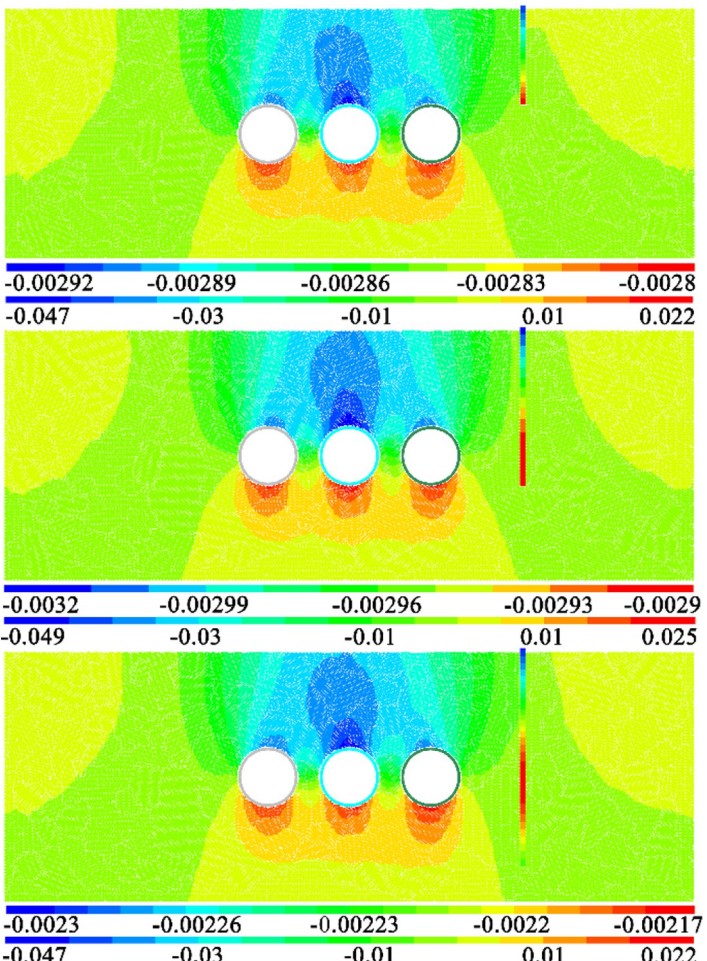

**Fig 7. Vertical displacement cloud map of three pile foundation models (m).**

construction is not significant. Fig 6 demonstrates that when the pile-tunnel distance is small, there is a slight difference in vertical displacement between the top and bottom of long piles, whereas the vertical displacement of the top and bottom of short piles is almost equal. This indicates that long piles undergo slight compressive deformation, while short piles experience overall settlement.

Fig 7 shows the contour of vertical displacement for pile and the surrounding soil when the pile-tunnel distance is 1D, for three different pile lengths. The results indicate that the vertical displacement at the pile top is greater than at the pile bottom. For short piles (when the pile length is less than H+D), the vertical displacement of the pile gradually decreases from the pile top to the pile bottom. However, for long piles, the vertical displacement is smallest at the tunnel depth and gradually increases below the tunnel depth.

Fig 8 shows the horizontal displacement of pile top and pile bottom of three different lengths of piles at different distances between tunnel and pile. In the figure, the positive value of horizontal displacement is the displacement away from the tunnel direction, and the negative value of horizontal displacement means that the pile foundation moves towards the tunnel. Fig 8 shows that the horizontal displacement of adjacent pile foundation induced by the construction of three-line small clear distance tunnel is much larger than the vertical displacement.

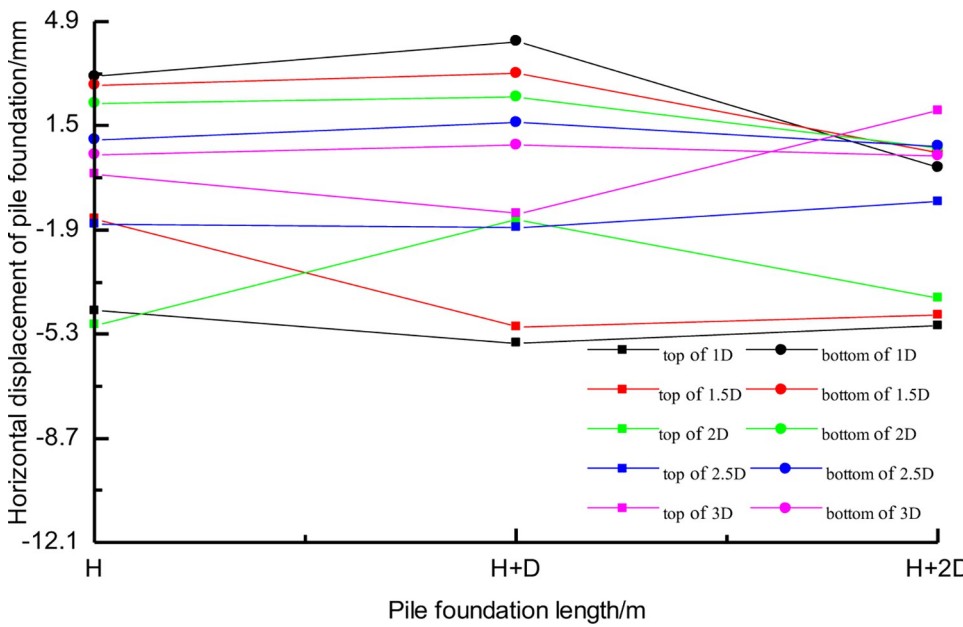

**Fig 8. Horizontal displacement of pile foundation with different pile lengths.**

When the distance between three kinds of pile foundation and tunnel is 3D, the horizontal displacement values of pile top and pile bottom are relatively close. However, when the distance between pile foundation and tunnel is 1D and 1.5D, the horizontal displacement difference between pile top and pile bottom caused by tunnel construction is large. When the distance between the tunnel and the pile is 1D, for the pile with the pile length H+D, the pile top moves 5.3mm towards the tunnel and the pile bottom moves 4.9mm away from the tunnel. The piles with length H+D are more sensitive to this effect, at this time the pile is mainly inclined to deformation, but when the pile length is H+2D there is a tendency of bending of the pile. For the cases set in this chapter, the pile foundation with pile length H+D is the most dangerous when the distance between the pile foundation and the tunnel is less than 1.5D [36].

Fig 9 shows the cloud map of horizontal displacement of strata and pile foundation caused by the construction of three-hole tunnel when the distance between tunnel and pile is 1D and three pile lengths. The first diagram is the horizontal displacement diagram of pile foundation, and the second diagram is the horizontal displacement diagram of stratum. Comparing and analyzing the horizontal displacement of the three pile length models, it can be found that the peak value of the horizontal displacement caused by the three tunnels is basically the same. The peak value of the negative direction of horizontal displacement is basically concentrated between 1cm and 1.2cm, and the peak value of the positive direction is basically concentrated between 1.2cm and 1.3cm. That is, the peak value of horizontal displacement in positive and negative directions is basically the same and the distribution law is roughly symmetric. In addition, it can also be found from the distribution law that the horizontal displacement caused by the construction of three tunnels shows four horizontal displacement concentration areas. The more significant positions are mainly concentrated on both sides of the tunnel and near the surface directly above the tunnel on both sides.

In addition, the horizontal displacement concentration area 2 in the figure has an impact on the pile tops of the three lengths of pile foundation, causing the pile tops to move towards the tunnel, which is also consistent with the law shown in Fig 9. The horizontal displacement

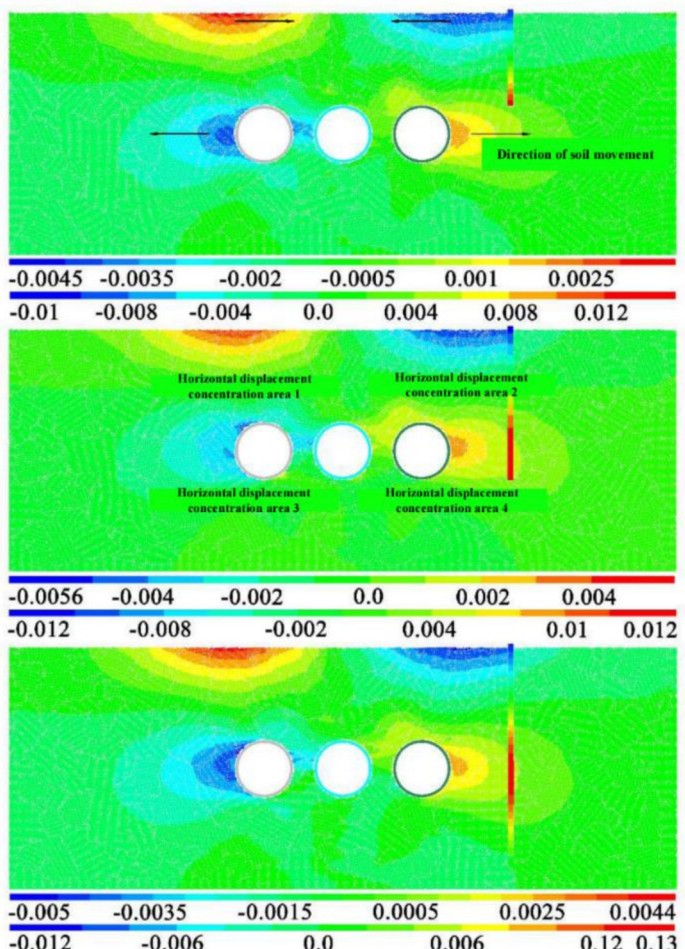

**Fig 9. Horizontal displacement cloud image at different pile lengths.**

concentration zone 4 mainly affects the pile foundation at one side of the tunnel. Therefore, for the two kinds of pile foundation with pile length H+D and H+2D, the construction of three-wire tunnel may cause the horizontal displacement response of the tunnel pile end or the pile foundation buried at the same depth as the tunnel.

In order to further analyze the distribution law of the influence of tunnel construction on the horizontal displacement of piles adjacent to the pile foundation, the horizontal displacement curves of three kinds of piles at different tunnel spacing were drawn, as shown in Fig 10. In the figure, the vertical distance of 0 represents the pile top, the buried depth of the tunnel is 10m, the tunnel diameter is 6.2m, that is, the vertical distance at the bottom of the tunnel is 16.2m.

In addition, the comparative analysis of the three pile lengths shows that the horizontal displacement value and the distribution of the horizontal displacement of the three pile lengths are basically the same near the buried depth of the tunnel [30]. However, the horizontal displacement at the top of pile shows greater difference with the increase of the distance between tunnel and pile foundation. For pile foundation with pile length H+2D, the horizontal displacement of pile bottom is not significant. This indicates that regardless of pile length, the influence of tunnel construction on pile foundation is mainly manifested in the height of horizontal displacement concentration area 2 and horizontal displacement concentration area 4 as shown in Fig 9.

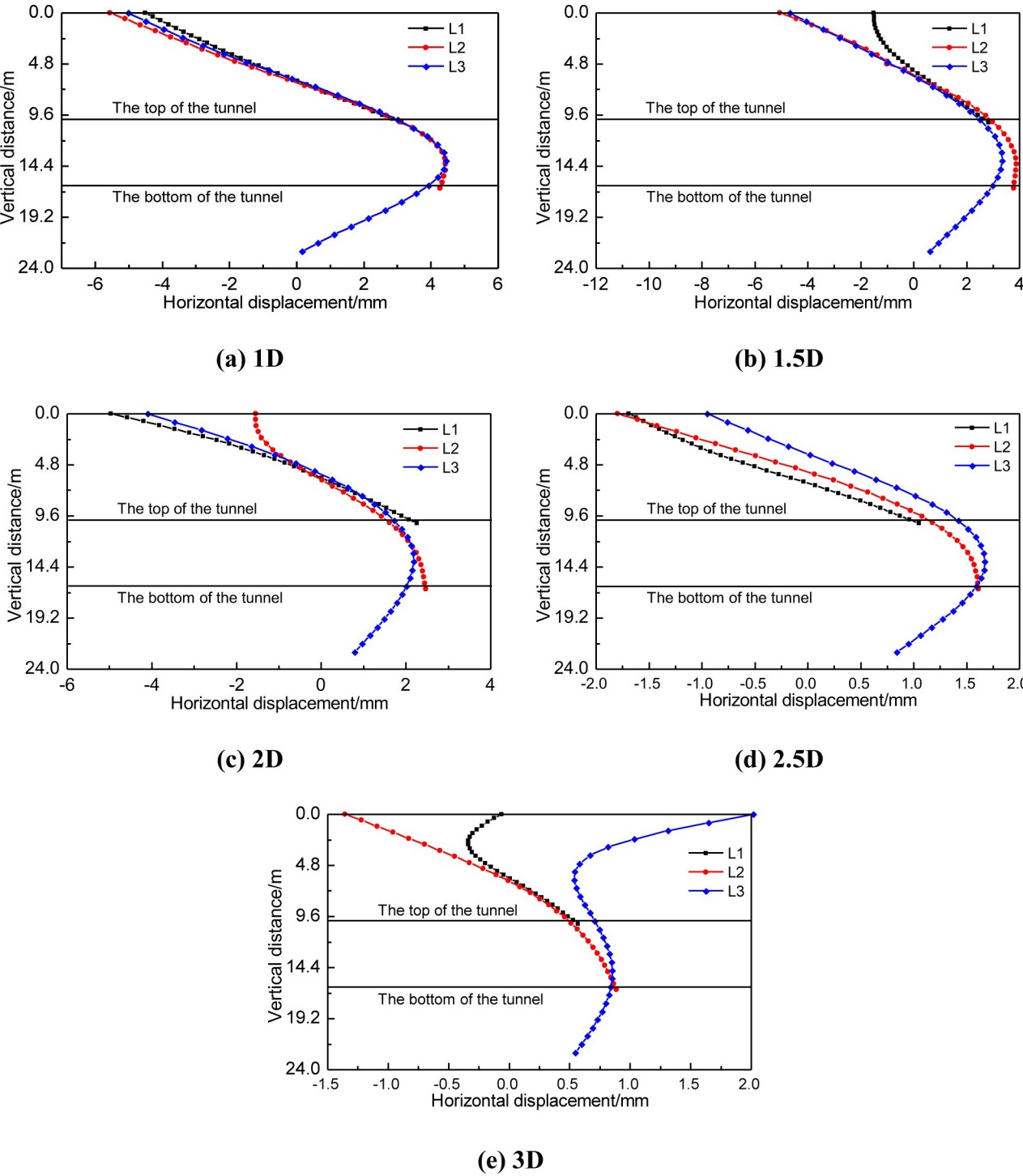

**Fig 10. Horizontal displacement of pile foundation at different distance from tunnel.**

The response of pile foundation to tunnel construction includes not only the displacement of pile body at different positions, but also the axial stress of pile foundation. Fig 11 shows the axial stress distribution of pile foundation with three lengths when the distance between tunnel and pile foundation is 1D, 2D and 3D respectively.

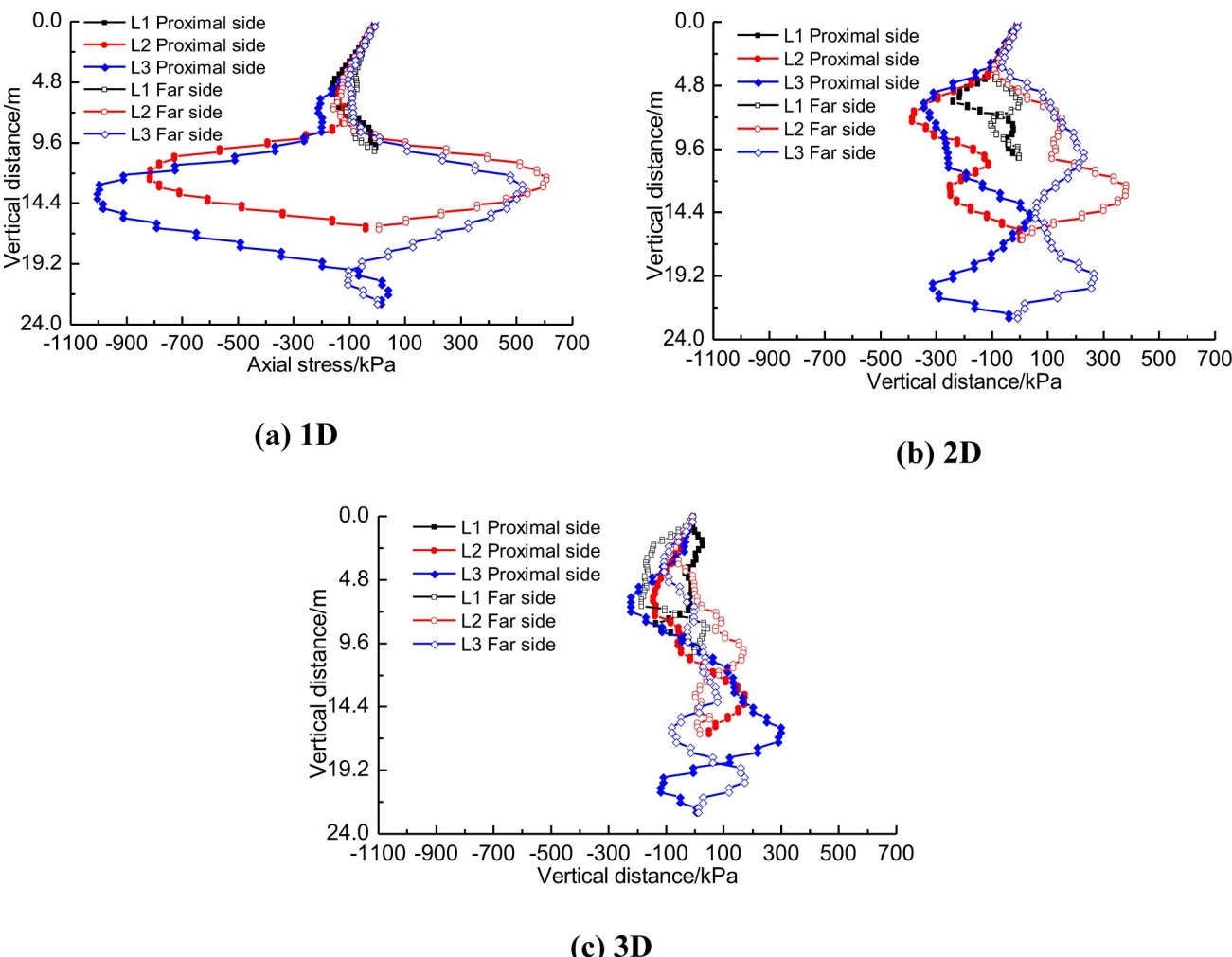

**(a) 1D**

**(b) 2D**

**(c) 3D**

**Fig 11. Axial stress distribution of three piles at different distances.**

As can be seen in Fig 11(A), when the distance between pile foundation and tunnel is 1D, long piles (pile length H+2D and H+D) mainly show bending. At this time, the axial stress near the tunnel side of the pile side reaches 1MPa, which is manifested as compressive stress, while the axial stress away from the tunnel side reaches 600kPa, which is manifested as tensile stress. This tensile stress could potentially cause cracking of the concrete protective layer on the surface of the steel bars, thereby increasing the risk of underground water corroding the reinforcement bars. Short piles (pile length H) are mainly characterized by inclination of the pile base and therefore axial stresses are low. a comparative analysis of Fig 11(B) to 11(C) also shows that as the distance between pile foundation and tunnel increases, the axial stress difference on both sides of pile foundation also gradually decreases, which also indicates that the influence of tunnel construction on pile foundation is gradually decreasing [30].

## Effect of pile-tunnel spacing

In this section, the deformation response of adjacent pile foundation caused by tunnel construction is calculated and analyzed under 5 kinds of spacing between pile foundation and tunnel. The influence of tunnel construction on adjacent pile foundation is mainly manifested as

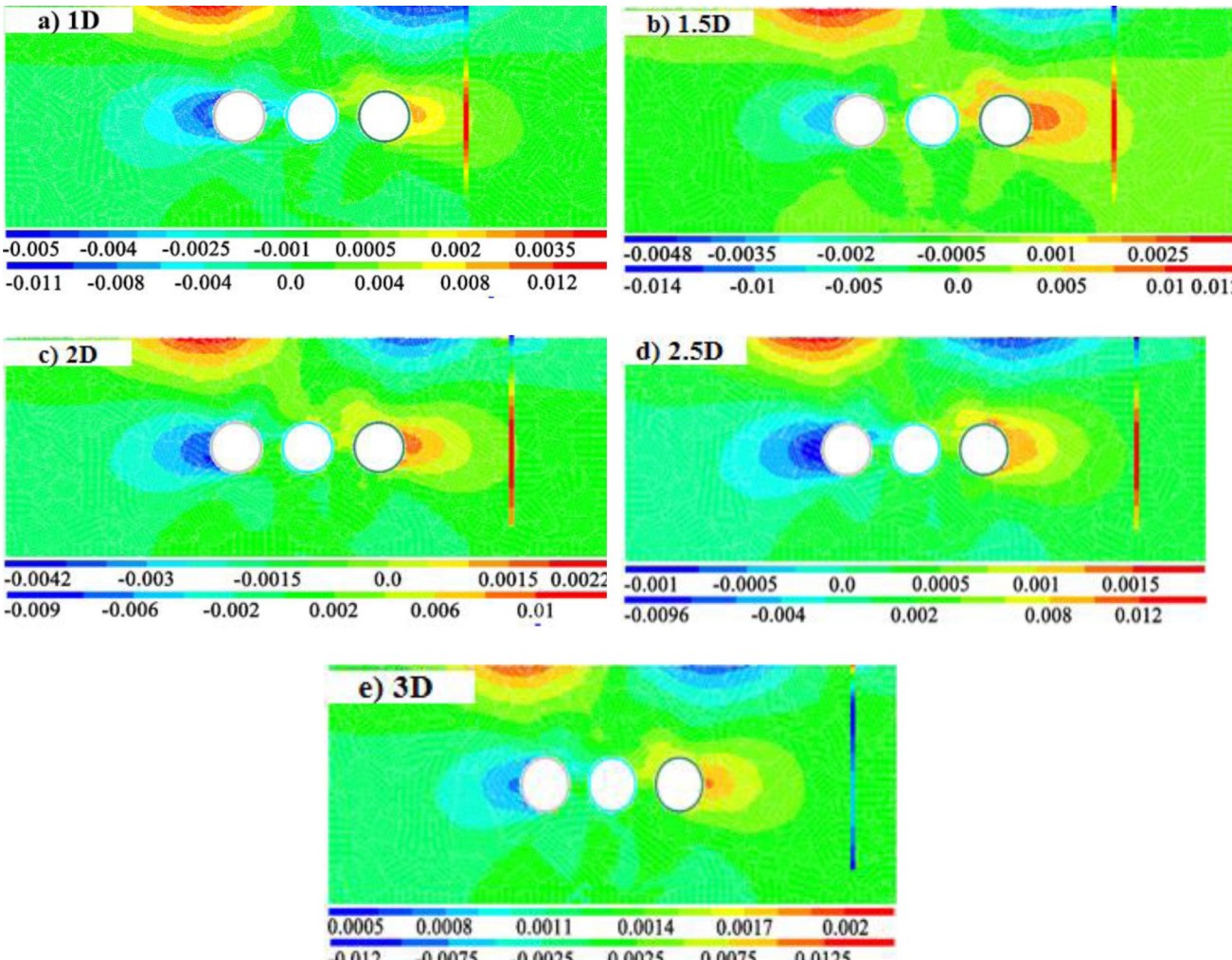

**Fig 12. Horizontal displacement cloud image when the distance between pile and tunnel is different.**

the horizontal deformation of pile foundation. By comparison and analysis of the five kinds of spacing in Fig 12, it can be found that the horizontal displacement of pile top begins to be less than 1mm after the distance between pile and tunnel is greater than 2.5D. At this time, the horizontal displacement of the pile body at the same position as the buried depth of the tunnel is only 1.5mm. It shows that when the distance between tunnel and pile is greater than 2.5D, the influence of tunnel construction on pile foundation is very weak [27].

Fig 13 shows the corresponding relationship between the horizontal displacement of pile top and pile bottom of three lengths and the distance between tunnel and pile foundation. It can be seen that as a whole, with the increase of the distance between pile foundation and tunnel, the horizontal displacement of pile top and pile bottom gradually decreases. There exists a difference in horizontal displacement between the pile top and bottom, which decreases as the distance between the pile and tunnel increases. There are some differences in the sensitivity of distance between tunnel and pile foundation. In contrast, the horizontal displacement of pile bottom shows a good linear correlation with the distance between tunnel and pile, in which the horizontal displacement of pilewith pile length H+2D is not affected by the distance

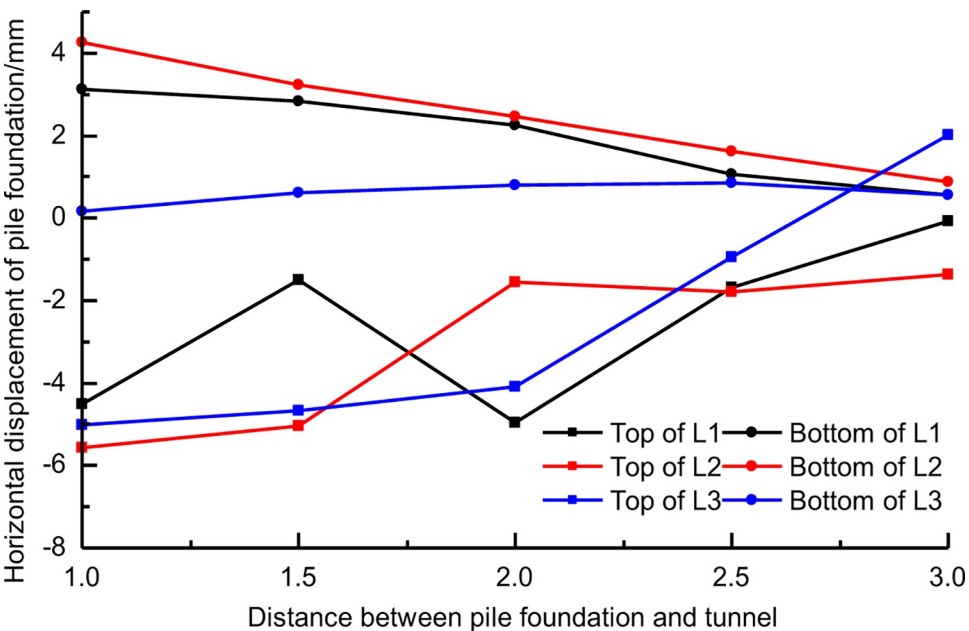

**Fig 13. Horizontal displacements of three kinds of piles with different spacing.**

between tunnel and pile, because the pile bottom of such long piles is no longer in the concentration area 4 of horizontal displacement caused by tunnel construction.

Fig 14 shows the distribution of horizontal displacements along the axis of piles with different distances between piles of three lengths and tunnels. The horizontal displacement of pile foundation with the same length can be compared and analyzed when the distance increases gradually. Fig 14(A) shows that, for short piles, the pile mainly inclines as the distance from the tunnel shrinks. When the distance between tunnel and pile is 1D, the horizontal displacement difference between pile top and pile bottom is 8mm.

The reason for this is that the disturbance caused by tunnel construction predominantly affects the soil mass located deep within and above the tunnel. The impact of tunnel construction on the soil mass below the depth of burial is relatively insignificant. Consequently, the soil mass beneath the depth of burial continues to constrain the deformation and displacement of the pile foundation during tunnel excavation.

The deep-buried soil mass within the tunnel experiences compression from the overlying soil mass and pipe segments, leading to a migration away from the tunnel's direction. This migration results in the bending of adjacent pile foundations. In a model extending H+2D in length, it becomes apparent that when the distance between the tunnel and the pile is 1D, the pile foundation showcases a displacement of 4.5mm away from the tunnel at the tunnel's depth. Simultaneously, the pile top displays a displacement of 5mm toward the tunnel. However, the pile's end doesn't exhibit significant horizontal displacement, yet all three segments distinctly exhibit bending. As the pile-tunnel spacing decreases, the tilting extent of short pile gradually increases, while the bending extent of long pile also gradually increases. For pile in service, the tilting of short pile is detrimental to the pile cap, whereas the bending of long pile induces additional axial stresses in the pile shaft.

Compared with the horizontal displacement, the vertical displacement of pile foundation has more significant correlation with the distance between tunnel and pile. Fig 15 shows the corresponding relationship between the vertical displacement of pile top and pile bottom and

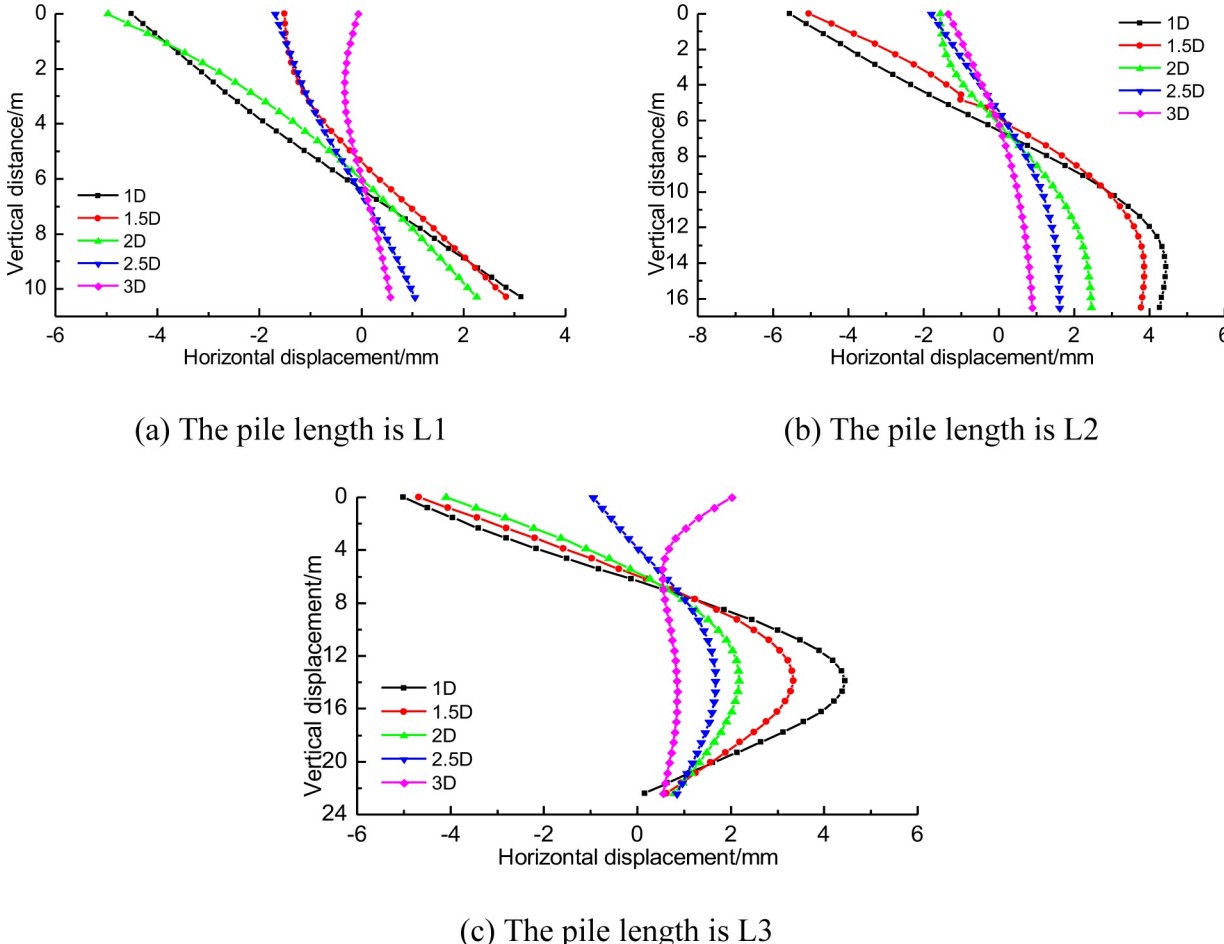

(a) The pile length is L1

(b) The pile length is L2

(c) The pile length is L3

**Fig 14. Distribution of horizontal displacements along the axis of piles at different distances.**

the distance between tunnel and pile foundation with three pile lengths. It can be found that the vertical displacement and spacing of pile foundation with three lengths show a good linear correlation within 3D. However, theoretically, with the further increase of the distance between tunnel and pile, the influence of tunnel construction on pile foundation should gradually weaken and disappear [32].

The construction of three-hole small distance tunnel will produce completely different stress response to pile foundation of different length. With the change of distance between tunnel and pile, the stress of pile body will also change in magnitude. There may even be changes in tension properties. Fig 16 shows the axial stress distribution of pile foundations of three lengths near the tunnel side. This section analyzes the axial stress distribution law of the side near the tunnel. Fig 16(A) shows that for short piles, the axial stress does not change significantly along the pile body because the pile foundation is mainly inclined as a whole. As the distance between pile foundation and tunnel shrinks, the axial stress of pile foundation does not show significant regularity. Fig 16(b) and 16(c) show that with the shortening of the distance between pile foundation and tunnel, greater axial stress will be generated at the depth of tunnel burial. According to Fig 16, it can be seen that this is obviously due to the increase of axial stress caused by the bending of pile foundation caused by tunnel construction.

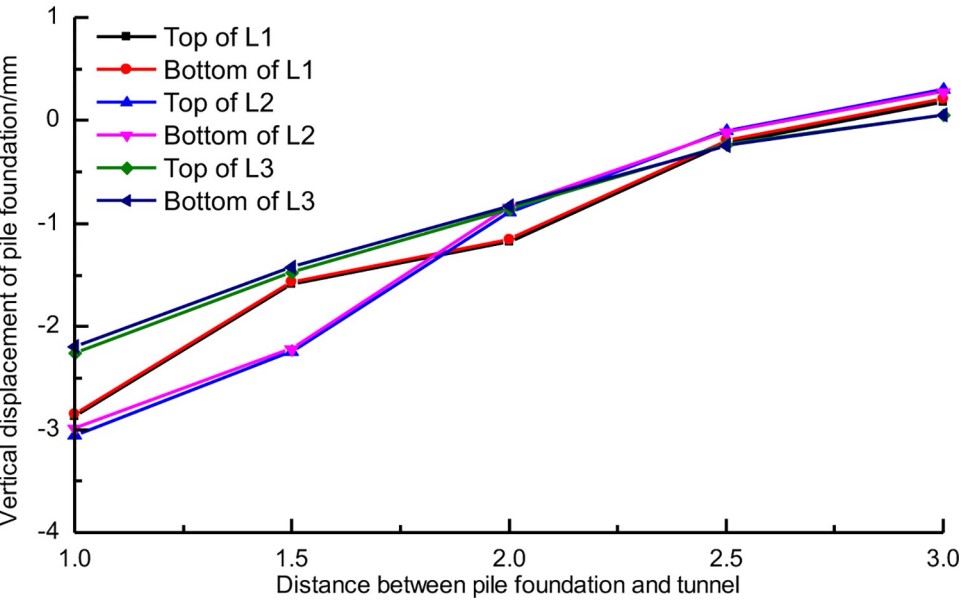

**Fig 15. Vertical displacement of three kinds of piles with different spacing.**

### Effect of three-line tunnel construction on pile

For the three-hole small distance tunnel, because the clear distance between the tunnels is smaller, there are repeated disturbance areas for soil layer. Therefore, with the gradual construction of the three lines, the influence on the adjacent pile foundation will be gradually significant. This section further analyzes the deformation and stress effects of the three-hole small distance tunnel on the adjacent pile foundation in different construction stages. Fig 17 shows the stratum and pile foundation horizontal displacement cloud map when the three lines are gradually excavated. Fig 17 shows the strata with five intervals and the horizontal displacement cloud map of pile foundation in the model with pile length H+2D. It can be seen that the horizontal displacement at the depth of the tunnel reached 1.6mm, while the peak value of the surface horizontal displacement reached 10-11mm. With the further construction of the third line, the distance between the tunnel and the pile is further narrowed, and the influence of tunnel construction on the pile foundation is further deepened. At this time, the pile foundation appears obvious bending, and the horizontal displacement at the depth of the tunnel reaches 3.5mm, while the displacement of the pile top toward the tunnel is 5mm, and the displacement of the pile bottom is still 0.

Fig 18 shows the horizontal displacement of pile top of three lengths of pile foundation at different excavation stages, and the horizontal coordinate is the three stages completed by three tunnels in turn. The horizontal displacement of pile top increases gradually with the three lines penetrating one by one. The farther the tunnel is from the pile foundation, the less significant the increment of pile top horizontal displacement caused by the three-hole tunnel. For pile foundation with pile length L2 (H+D), when the distance between pile and tunnel is 3D, the horizontal displacement increment of pile top is less than 1.6mm with the penetration of three tunnels one by one. For pile foundation with pile length L3(H+2D), when the distance between pile and tunnel is 2.5D and 3D, the incremental horizontal displacement of pile top caused by double hole penetration is less than 0.5mm. However, when the three holes are connected, the incremental horizontal displacement of pile top is relatively larger, about 1–1.5mm, which is also reflected in Fig 18(A) and 18(B).

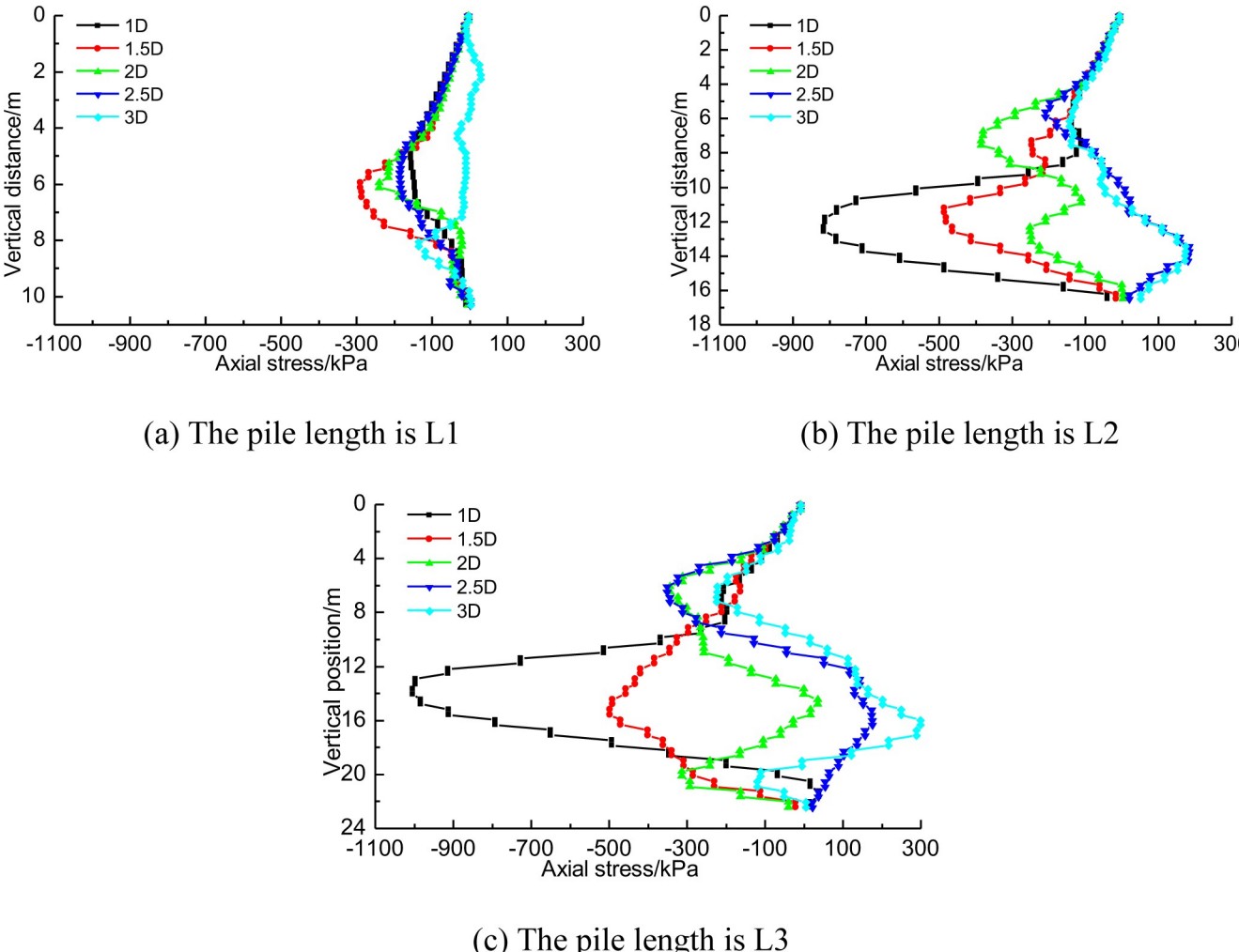

(a) The pile length is L1

(b) The pile length is L2

(c) The pile length is L3

**Fig 16. Near-side axial stress difference of three kinds of piles at different distances.**

Fig 19 shows the horizontal displacement of pile bottom for three lengths of pile foundation at different excavation stages. The horizontal positions of the pile bottoms of the three kinds of piles do not show significant positive correlation with the three kinds of tunnels penetrating one by one. The reason is that when only two lines are connected, the pile foundation is relatively in the horizontal displacement concentration area on both sides of the tunnel. However, when the third line is through, if the distance between pile foundation and tunnel is very close, the position of pile bottom is below the horizontal displacement concentration area on both sides of the tunnel. Therefore, the horizontal displacement does not increase, and even decreases [34]. For long pile at small spacing, the horizontal displacement at the pile bottom first increases and then decreases as the three tunnels are sequentially completed. This phenomenon becomes less noticeable as the spacing increases.

The horizontal displacement and vertical displacement of the adjacent pile foundation have corresponding changes, so the axial stress will also change. Fig 20 shows the change law of axial stress of pile foundation near the tunnel side with the one-by-one penetration of three lines.

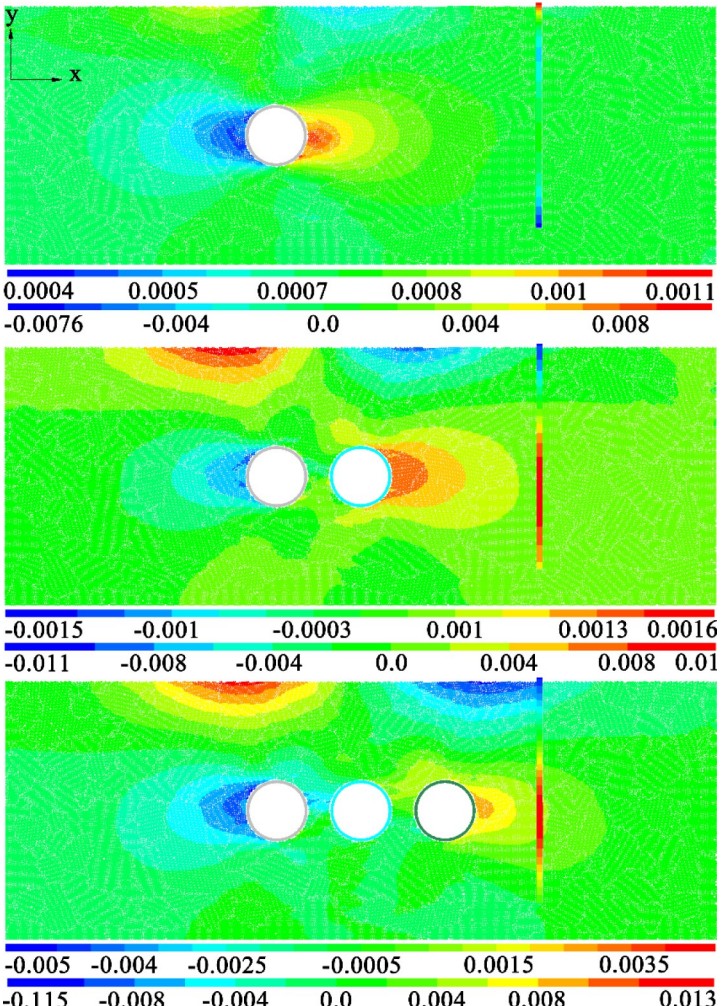

**Fig 17. Cloud map of horizontal displacement of adjacent pile foundation and stratum induced by tunnel construction.**

For short piles with length H, no matter the distance between the piles and the tunnels is 1D or 3D, the axial stress on one side of pile foundation near the tunnels caused by the construction of three tunnels is not significant. When the distance between the tunnel and pile foundation is the same, the long pile shows stronger sensitivity to the one-by-one penetration of the three-hole tunnel with small distance. For example, in Fig 20(C), when the construction of tunnel 1 was completed, the axial stress distribution of pile foundation near the tunnel side ranged from -100kPa to 200kPa. However, when the construction of tunnel 3 is completed, the axial stress distribution range near the tunnel side of pile foundation reaches -1000kPa to 200kPa. By comparing Fig 20(B), 20(E) and 20(H), it can be found that for the same pile length, with the increase of the distance between the tunnel and the pile foundation, the sensitivity of the axial stress near the tunnel of the pile foundation to the construction of the three-wire tunnel gradually decreases. When the distance between the tunnel and the pile foundation is 1D, for the pile foundation with the length H+D, the axial stress near the tunnel side of the pile foundation generated after the construction of tunnel 1 is concentrated between -300kPa and 200kPa. When the construction of the third line is completed, the axial stress of the pile foundation near the tunnel side is concentrated between -850kPa and 200kPa.

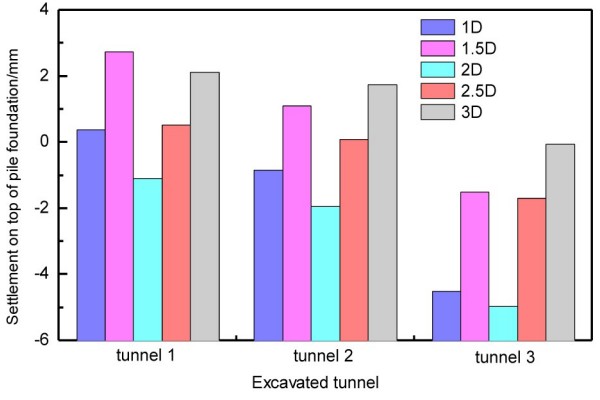

(a) The pile length is L1

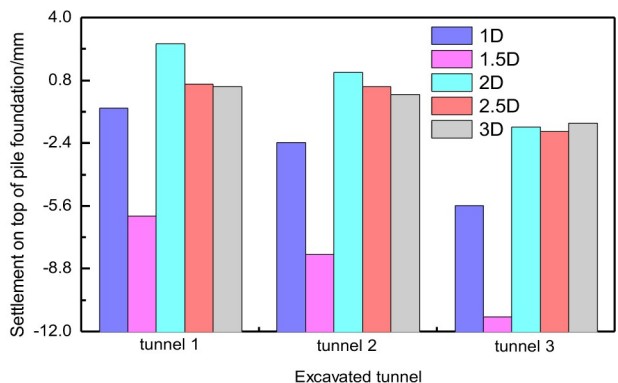

(b) The pile length is L2

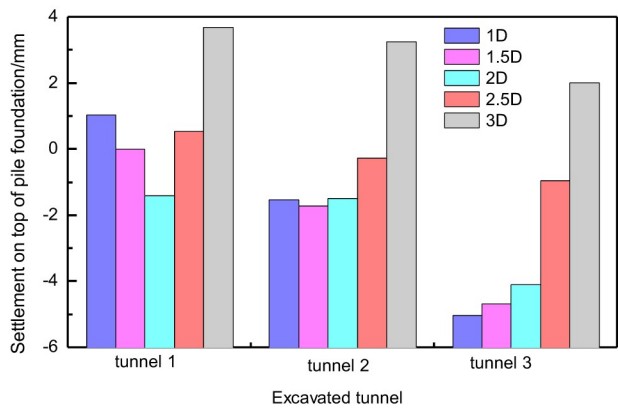

(c) The pile length is L3

**Fig 18. Horizontal displacement of pile top of three kinds of piles at different excavation stages.**

### Mitigation measures and design considerations

The vibration and deformation of pile foundation caused by tunnel construction may affect the structural stability and soil mechanical properties. These impacts may require a range of measures to manage and mitigate before, during and after construction. The foregoing study discussed the pile foundation response of the three-cave tunnel under different conditions. This section will take relevant mitigation measures to cause stratum disturbance and soil deformation of the three-cave tunnel, so as to reduce the impact on the pile foundation, and put forward mitigation suggestions.

1. Geological investigation and analysis: carry out detailed geological investigation before construction to understand the formation conditions and geological characteristics. This helps to predict possible geological changes and the impact of tunnel construction on the soil layer.

2. Structural design optimization: By optimizing the structural design of the tunnel, the impact on the surrounding soil and pile foundation can be reduced. This may include adjusting the depth, width and shape of the tunnel to reduce formation disturbance.

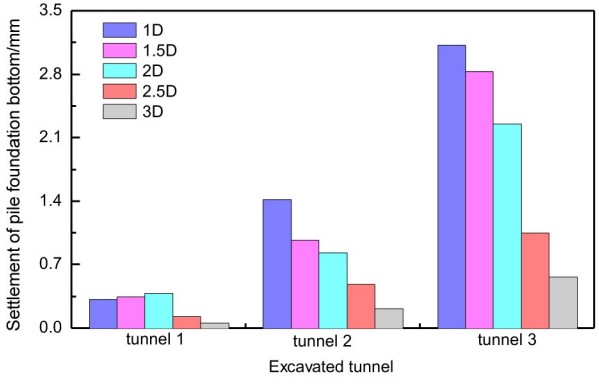

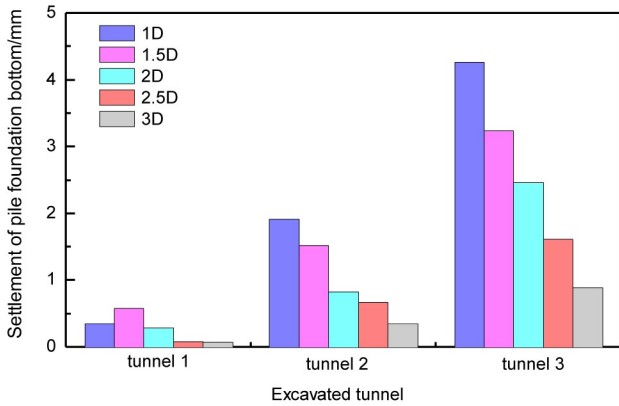

(a) The pile length is L1                                    (b) The pile length is L2

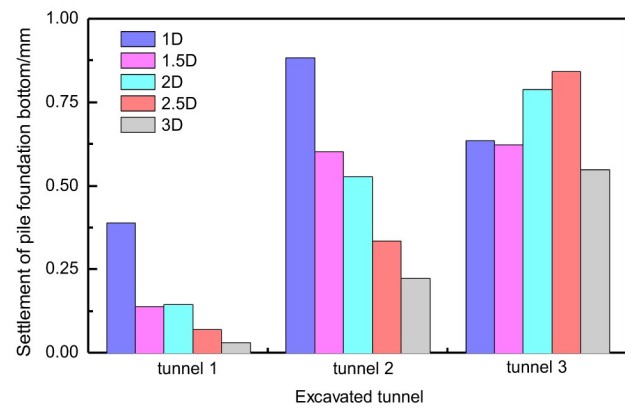

(c) The pile length is L3

**Fig 19. Horizontal displacement of pile bottom of three kinds of piles at different excavation stages.**

3. Use appropriate construction techniques: Grouting is a critical process in shield tunneling that efficiently fills gaps behind the shield. The deformation of pile induced by tunnel excavation is primarily horizontal at the depth of the tunnel. Therefore, adjusting the grouting pressure and volume on the side of the tunnel facing the piles can reduce additional displacement of the pile.

4. Formation reinforcement and support measures: Take formation reinforcement and support measures around the tunnel, such as the use of supporting piles, soil reinforcement or other geological engineering techniques to protect the pile foundation from disturbance.

5. Monitoring and adjustment: real-time monitoring during the construction process, and adjustment and control according to the monitoring results. This helps to identify problems in a timely manner and take necessary corrective actions to reduce negative impacts.

The above methods are helpful to reduce the negative effects of three-hole tunnel construction on pile foundation. Considering the engineering practice, technical means and management strategy comprehensively, the disturbance of tunnel construction on adjacent pile

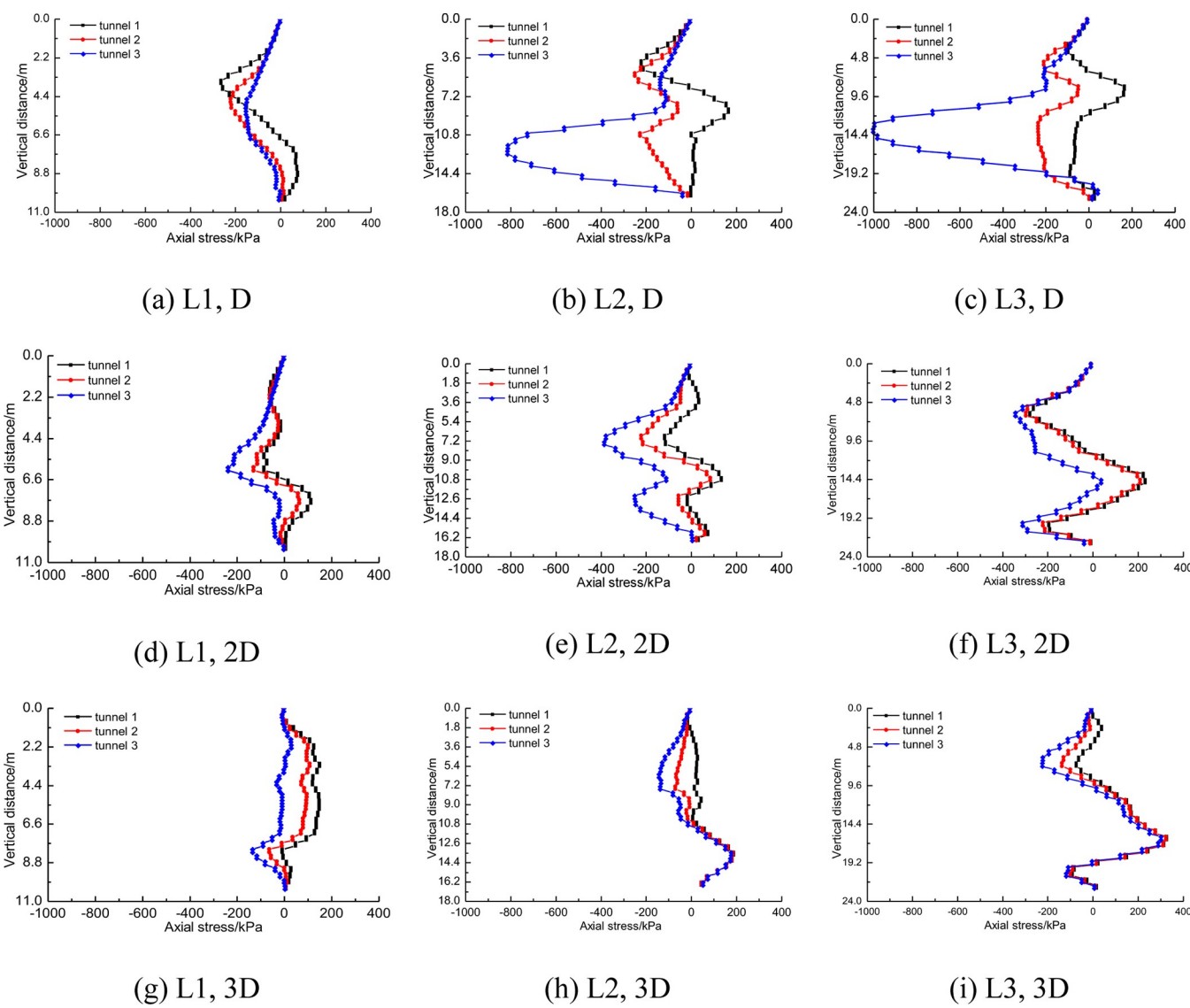

**Fig 20. Influence of three-hole tunnel construction on axial stress of pile foundation.**

foundation can be effectively alleviated. In order to ensure engineering safety and structural stability, it is necessary to use various measures to minimize the adverse effects. In the actual project, it is necessary to consider the geological conditions, construction technology and design scheme to develop the best mitigation measures to reduce the adverse impact on the pile foundation and the surrounding soil.

## Conclusions

In this paper, the DEM-FDM coupling method was used to analyze the deformation response of pile foundation caused by three-hole tunnel excavation. The influence of the length of the pile foundation, the distance between the tunnel and pile foundation, and construction factors on displacement and stress were discussed. The conclusions are as follows:

The impact of tunnel construction on the horizontal displacement of pile is significant, primarily concentrated at the depth of the tunnel. During tunnel construction, short pile tend to

tilt, while long pile exhibit a bending trend. Pile with a length of H+D are most sensitive to tunnel construction. The axial stress of short pile shows little change when they tilt, whereas the axial stress of long pile increases when they bend.

The influence range of tunnel construction on pile is approximately 2.5D. Within a range of 3D, the horizontal displacement of the pile top and bottom shows a linear correlation with the pile-tunnel distance. A decrease in the pile-tunnel distance causes long pile to bend. When the pile-tunnel distance is 1D and the pile length is H+2D, the peak axial stress of the tunnel reaches 1MPa.

At small pile-tunnel distances, the horizontal displacement of the pile top gradually decreases as the three tunnels are sequentially constructed, with the maximum displacement occurring after the completion of the first tunnel. The horizontal displacement at the bottom of short piles gradually increases, while that of long pile first increases and then decreases. The axial stress of long pile is more sensitive to the repeated disturbances of the three-tunnel construction.

This paper puts forward some meaningful conclusions, however, there are still some limitations. Firstly, the coupling models in this paper are all 2D methods. Although 2D methods can simulate deformation on a plane, in reality, the tunnelling process is a three-dimensional problem. In the future, researchers can use FLAC3D and PFC3D for coupling simulations. Secondly, synchronous grouting is not considered in the simulation process of this paper. Although the selection of tunnel excavation diameter and segment outside diameter in this paper is subject to the actual situation, there will be some deviation in the simulation results. In the future research, the numerical simulation of synchronous grouting process can be considered to improve the accuracy of simulation results. Finally, in the future, researchers can supplement the effects of soil properties, groundwater conditions, and tunnel construction methods to improve the depth and breadth of research.

## Supporting information

**S1 File. The results of numerical simulation verification.**
(XLSX)

## Acknowledgments

We would like to acknowledge the reviewers and editors for their valuable comments and suggestions.

## Author Contributions

**Data curation:** Yuan Zhang.

**Formal analysis:** Yuan Zhang.

**Investigation:** Zhenchu Zhao.

**Methodology:** Yuan Zhang, Fang Dai.

**Project administration:** Yuan Zhang.

**Software:** Fang Dai.

**Supervision:** Zhenchu Zhao.

**Validation:** Zhenchu Zhao.

**Visualization:** Fang Dai.

Writing – original draft: Zhenchu Zhao.

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
