## [Decision Letter · Decision Letter 0]

26 Feb 2024

PONE-D-24-03222Study on deformation of tunnel pile foundation based on DEM and FDM couplingPLOS ONE

Dear Dr. Dai,

Thank you for submitting your manuscript to PLOS ONE. After careful consideration, we feel that it has merit but does not fully meet PLOS ONE’s publication criteria as it currently stands. Therefore, we invite you to submit a revised version of the manuscript that addresses the points raised during the review process.

We look forward to receiving your revised manuscript.

Kind regards,

Mohammad Azadi

Academic Editor

PLOS ONE

Journal Requirements:

"The author(s) received no specific funding for this work"

4. We note that your Data Availability Statement is currently as follows: [All relevant data are within the manuscript and its Supporting Information files]

Additional Editor Comments:

The manuscript must be revised based on the reviewers’ comments plus the following issues,

1) A separated file must be provided for the authors’ answers to the comments, one by one. Moreover, all changes must be yellow-colored highlighted sentences in the revised article. The track changes condition is not suggested.

2) No abbreviations must be used in the title.

3) All abbreviations should be defined at first mentioning, such as DEM-FDM, etc.

4) No quantitative data could be seen in the abstract.

5) All keywords must be found in the abstract or the title.

6) The introduction is lengthy. Moreover, the novelty must be highlighted in the introduction, compared to the literature review.

7) All used material properties need references, especially for the data in Tables 1 and 2.

8) The structure is confusing. The text must have an introduction, research method, results and discussion, conclusions, and references.

9) Numerical results must be validated such as Figures 6 and 8.

10) "Conclusion" must be changed to "Conclusions". It is also lengthy. It must be shortened.

11) The discussion is poor and it must be improved. They must be compared to other results of other similar articles.

12) References should be updated based on recent articles, published in 2015-2024. Moreover, it should be extended to at least 35 articles for a proper discussion.

Reviewers' comments:

Reviewer's Responses to Questions

**Comments to the Author**

1. Is the manuscript technically sound, and do the data support the conclusions?

Reviewer #1: Yes

Reviewer #2: Partly

2. Has the statistical analysis been performed appropriately and rigorously? 

Reviewer #1: Yes

Reviewer #2: N/A

3. Have the authors made all data underlying the findings in their manuscript fully available?

Reviewer #1: Yes

Reviewer #2: Yes

4. Is the manuscript presented in an intelligible fashion and written in standard English?

Reviewer #1: Yes

Reviewer #2: No

5. Review Comments to the Author

Reviewer #1: This research paper addresses an important issue in engineering, specifically the deformation of pile foundations due to shield tunnel construction. The authors propose using the DEM-FDM coupling numerical simulation method to study the deformation response mechanism. This method appears to combine the discrete element method (DEM) and the finite difference method (FDM) to simulate the deformation process of pile foundations during tunnel driving. The paper focuses on three factors: the length of the pile foundation, the distance between the tunnel and the pile foundation, and the small distance between the three holes. The results indicate that the displacement and deformation response of adjacent pile foundations due to the construction of a three-hole tunnel with small spacing primarily involve horizontal displacement, bending, and inclination of the pile foundation.

Overall, this paper presents a valuable contribution to the field of geotechnical engineering but could benefit from further elaboration and contextualization. Hence, I recommend minor revision before acceptance in the Journal.

1.Did the researchers conduct any experimental validation or comparison with field data to verify the accuracy of their simulation results? It is suggested that researchers supplement or explain the comparative verification content.

2.Given the formation disturbance and horizontal soil deformation caused by the construction of the three-hole tunnel, are there any mitigation measures or design considerations proposed to minimize the negative effects on pile foundations?

3.The results describe the displacement and deformation response of pile foundation under the influence of three-hole and small-distance tunnel construction, and the deformation response under the influence of three factors. However, when describing the results, it is recommended that more quantitative analysis be provided to support the accuracy and credibility of the conclusions.

4.This paper clearly points out the purpose of the study, that is, to investigate the deformation response mechanism of tunnel pile foundation caused by shield construction, and describes the numerical simulation method of DEM-FDM coupling. It is recommended to provide a more detailed introduction in the method section, including the steps of numerical simulation, parameter setting and model validation, so as to ensure that readers can fully understand the reliability and applicability of the research method.

The readability of the manuscript is generally ok, but an editorial review by a native speaker must be conducted in places throughout the document.

Reviewer #2: This paper investigates the deformation response mechanism of pile foundation induced by the construction of three-hole small clear distance tunnel, by use of the DEM-FDM coupled numerical simulation method. Three factors, i.e., the length of pile foundation, the distance between tunnel and pile foundation, and the small clear distance of three-hole tunnel construction, have been focused on in this study. Some valuable results have been yielded; yet, the following issues should be clarified before it can be accepted for publication.

(1) The deformation study of adjacent pile foundation is mainly aimed at the deformation response induced by the construction of single tunnel or double tunnel. This study focuses the action mechanism and deformation response of the three-hole small distance tunnel construction to the pile foundation of adjacent buildings. Can authors provide some engineering background for the three-hole small distance tunnel construction near pile foundation?

(2) For the DEM part, calibration process of the soil parameters (for example, against the results from the triaxial tests) should be given.

(3) In fact, DEM is not an appropriated numerical tool for modelling the process of tunnelling, given that it cannot mimicking simultaneous grouting. The soil loss rate is 6.15% based on the diameter of the tunnel excavation and the outer diameter of the segment. But adoption of this value will overestimate the disturbance tunnelling to the surrounding soil and pile foundation, as simultaneous grouting was not considered in the simulation.

(4) The paper used FLAC2D and PFC2D to simulate the deformation process of pile foundation during tunnel driving. The tunnelling problem actually is an 3D problem; therefore, the limitation of the study should be highlighted in the discussions and conclusions.

6. PLOS authors have the option to publish the peer review history of their article (what does this mean?). If published, this will include your full peer review and any attached files.

Reviewer #1: **Yes: **Yunhao Chen

Reviewer #2: No

---

## [Author Response · Author response to Decision Letter 0]

19 Mar 2024

Dear Editors and Reviewers:

We sincerely appreciate the valuable feedback from the editor and all the reviewers, which has helped us improve the quality of the manuscript. The questions are answered as follows:

Editor Comments:

1) A separated file must be provided for the authors’ answers to the comments, one by one. Moreover, all changes must be yellow-colored highlighted sentences in the revised article. The track changes condition is not suggested.

Response：

All comments have been revised one by one according to the editor's opinion. Highlight the revised manuscript in yellow.

2) No abbreviations must be used in the title.

Response：

Modify the title to“Study on deformation of tunnel pile foundation based on discrete element method and finite difference method coupling”

3) All abbreviations should be defined at first mentioning, such as DEM-FDM, etc.

Response：

All abbreviations are defined at the first time when the article appears, for example, DEM-FDM has been defined in the abstract line 10 and the body line 68.

4) No quantitative data could be seen in the abstract.

Response：

Some expressions in the abstract were modified and the quantitative data were supplemented. The supplement has been highlighted in yellow.

5) All keywords must be found in the abstract or the title.

Response：

It has been verified that all keywords can be found in the summary or title.

6) The introduction is lengthy. Moreover, the novelty must be highlighted in the introduction, compared to the literature review

Response：

At the same time, the method innovation of this paper has been explained in the introduction, and the number of references has been supplemented. The changes have been highlighted in yellow.

7) All used material properties need references, especially for the data in Tables 1 and 2.

Response：

The data sources in Table 1 and Table 2 are laboratory experiments. As for the experimental sources of materials, this part has been supplemented in Chapter 2 of the paper.

8) The structure is confusing. The text must have an introduction, research method, results and discussion, conclusions, and references.

Response：

The article structure is adjusted from introduction, research method, results and discussion, conclusions, and references. Five parts have been described.

9) Numerical results must be validated such as Figures 6 and 8.

Response：

Since the displacement of pile foundation is difficult to collect in actual monitoring, this paper compares the actual monitoring of other collected data with the numerical simulation, so as to illustrate the reliability of the numerical simulation in this paper.

In this paper, the comparison of surface settlement curves has been used to verify the numerical simulation. The detailed verification content can be found from line234 to 246 of the revised draft.

10) Conclusion must be changed to Conclusions. It is also lengthy. It must be shortened.

Response：

According to the editor's opinion, the conclusions of the article have been simplified. 

11) The discussion is poor and it must be improved. They must be compared to other results of other similar articles.

Response：

According to the editor's opinion, the discussion of the article has been simplified. In addition to this, comparisons with other articles have been added.

12) References should be updated based on recent articles, published in 2015-2024. Moreover, it should be extended to at least 35 articles for a proper discussion.

Response：

The references of the article have been supplemented to 36 articles.

 

Reviewer 1:

This research paper addresses an important issue in engineering, specifically the deformation of pile foundations due to shield tunnel construction. The authors propose using the DEM-FDM coupling numerical simulation method to study the deformation response mechanism. This method appears to combine the discrete element method (DEM) and the finite difference method (FDM) to simulate the deformation process of pile foundations during tunnel driving. The paper focuses on three factors: the length of the pile foundation, the distance between the tunnel and the pile foundation, and the small distance between the three holes. The results indicate that the displacement and deformation response of adjacent pile foundations due to the construction of a three-hole tunnel with small spacing primarily involve horizontal displacement, bending, and inclination of the pile foundation.

Overall, this paper presents a valuable contribution to the field of geotechnical engineering but could benefit from further elaboration and contextualization.

1. Did the researchers conduct any experimental validation or comparison with field data to verify the accuracy of their simulation results? It is suggested that researchers supplement or explain the comparative verification content.?

Response：

Thank you very much for the valuable suggestions of the reviewers. The data sources of the numerical simulation in this paper mainly include the following three parts:

（1）Laboratory tests: In the DEM field, the contact parameters of particles are calibrated through a series of large-scale triaxial test analyses. In the process of numerical simulation in this paper, the selection of parameters is subject to field monitoring, and the calibration test is carried out by triaxial test, and the modeling parameters of the following soils are obtained. Triaxial test is a common method to determine the macroscopic parameters of soil. Unconsolidated and undrained test is carried out on a series of sand and clay with density gradient through the triaxial tester. The relationship between the internal friction Angle ϕ, cohesion force c, deformation modulus k and porosity is fitted, and the correlation between the shear strength and porosity of the soil is studied, which provides a basis for the parameter assignment of the subsequent shield model.

（2）Field observation and monitoring: This paper uses field observation and monitoring data to verify the accuracy of numerical simulation. This paper collects the actual data of the tunnel construction site, such as the change of ground settlement, pile foundation deformation, etc., and compares these data with the simulation results. The following is a comparison between the results of numerical simulation of land surface settlement change and the results of field monitoring. As can be seen from the figure below, the numerical model in this paper has a certain reliability, and the verification and comparison of the surface settlement values indicate that the model has a strong fitting ability.

 （3） Comparison of case studies: By comparing data from previous case studies of similar projects, researchers can assess the reliability of numerical simulation results. The above three comparison methods can verify the reliability of the numerical simulation.

2. Given the formation disturbance and horizontal soil deformation caused by the construction of the three-hole tunnel, are there any mitigation measures or design considerations proposed to minimize the negative effects on pile foundations?

Response：

According to the opinions of the experts, in the conclusion part of the paper, the related mitigation measures caused by the three-hole tunnel are added to reduce the impact on the pile foundation, and the mitigation suggestions are put forward. At the same time, the discussion part of Section 4.4 is added to the revised manuscript to explain the mitigation of disturbance effects of tunnel construction on adjacent pile foundations. The specific content is elaborated in lines 494 to 522 of the revised manuscript, which is as follows:

（1） Geological investigation and analysis: carry out detailed geological investigation before construction to understand the formation conditions and geological characteristics. This helps to predict possible geological changes and the impact of tunnel construction on the soil layer.

（2）Structural design optimization: By optimizing the structural design of the tunnel, the impact on the surrounding soil and pile foundation can be reduced. This may include adjusting the depth, width and shape of the tunnel to reduce formation disturbance.

（3） Use appropriate construction techniques: Select appropriate construction techniques and methods, such as shield tunneling or other methods to reduce

（4）Formation reinforcement and support measures: Take formation reinforcement and support measures around the tunnel, such as the use of supporting piles, soil reinforcement or other geological engineering techniques to protect the pile foundation from disturbance.

（5） Monitoring and adjustment: real-time monitoring during the construction process, and adjustment and control according to the monitoring results. This helps to identify problems in a timely manner and take necessary corrective actions to reduce negative impacts.

The above methods are helpful to reduce the negative effects of three-hole tunnel construction on pile foundation. In the actual project, it is necessary to consider the geological conditions, construction technology and design scheme to develop the best mitigation measures to reduce the adverse impact on the pile foundation and the surrounding soil.

3. The results describe the displacement and deformation response of pile foundation under the influence of three-hole and small-distance tunnel construction, and the deformation response under the influence of three factors. However, when describing the results, it is recommended that more quantitative analysis be provided to support the accuracy and credibility of the conclusions.

Response：

According to the expert opinion, the conclusion part of the article has been modified and simplified. The quantitative analysis under different factors is supplemented, and the revised conclusions are as follows:

In this paper, the DEM-FDM coupling method was used to analyze the deformation response of pile foundation caused by three-hole tunnel excavation. The influence of the length of the pile foundation, the distance between the tunnel and pile foundation, and construction factors on displacement and stress were discussed. The conclusions are as follows:

(1) Influence on pile foundations of different lengths: It was found that when the distance between the pile foundation and tunnel was 3D, tunnel construction did not cause vertical displacement of the pile foundation. The difference in extreme vertical deformation caused by tunnel construction was not significant, and settlement displacement was concentrated in the range of 4.7cm to 4.9cm. The peak value of uplift was concentrated between 2.2cm and 2.5cm. The actual monitoring results and numerical simulation results were mutually verified.

(2) Influence on the distance between the tunnel and pile foundation: It was found that as the distance between the pile foundation and tunnel decrease, the pile mainly tilted. When the distance between the tunnel and pile foundation was 1D, the horizontal displacement difference between pile top and pile bottom was 8mm. It was found that the three lengths of the pile foundation showed a good linear correlation within 3D. However, theoretically, with the further increase in the distance between the tunnel and pile, the influence of tunnel construction on the pile foundation should gradually weaken and disappear.

(3) The influence of three-hole tunnel construction on pile foundation: It was found that the horizontal displacement of the pile top increased gradually with the penetration of three tunnels one by one. For H+2D pile foundation, the incremental horizontal displacement of the pile top caused by double-hole penetration was less than 0.5mm. However, when the three-hole tunnel was connected, the horizontal displacement increment of the pile top was relatively larger, about 1.5mm.

4. This paper clearly points out the purpose of the study, that is, to investigate the deformation response mechanism of tunnel pile foundation caused by shield construction, and describes the numerical simulation method of DEM-FDM coupling. It is recommended to provide a more detailed introduction in the method section, including the steps of numerical simulation, parameter setting and model validation, so as to ensure that readers can fully understand the reliability and applicability of the research method.

Response：

Detailed steps for additional numerical simulation and parameter setting, based on expert advice, can be found in line 164 -170 of the revised manuscript.

5. The readability of the manuscript is generally ok, but an editorial review by a native speaker must be conducted in places throughout the document.

Response：

Thank you very much for your advice. We have carefully checked the entire article to avoid spelling, grammar and typographical errors in order to improve the readability and language quality of the article. Grammar or presentation issues that have been revised in the text are corrected in red in the revised manuscript. Thank you for your guidance and support!

Reviewer 2:

This paper investigates the deformation response mechanism of pile foundation induced by the construction of three-hole small clear distance tunnel, by use of the DEM-FDM coupled numerical simulation method. Three factors, i.e., the length of pile foundation, the distance between tunnel and pile foundation, and the small clear distance of three-hole tunnel construction, have been focused on in this study. Some valuable results have been yielded; yet, the following issues should be clarified before it can be accepted for publication.

1. The deformation study of adjacent pile foundation is mainly aimed at the deformation response induced by the construction of single tunnel or double tunnel. This study focuses the action mechanism and deformation response of the three-hole small distance tunnel construction to the pile foundation of adjacent buildings. Can authors provide some engineering background for the three-hole small distance tunnel construction near pile foundation?

Response：

According to the expert opinion, this paper adds the engineering background of three-hole small-distance tunnel construction near pile foundation. This paper is based on a shield section of Nantong rail Transit Line 1. The section of the shield tunnel runs along the existing road. Along the line are the city's main roads and important commercial blocks and residential areas. The underground pipeline along the line is dense and passes through the existing civil air defense tunnel. The buildings and roads traversed by the shield section are shown in the figure below, which are supplemented in the line211-218 of the article.

2. For the DEM part, calibration process of the soil parameters (for example, against the results from the triaxial tests) should be given.

Response：

According to the expert opinion, the soil parameter test process of the paper is supplemented, and the specific content is as follows.

In the DEM field, the contact parameters of particles are calibrated through a series of large-scale triaxial test analyses. In the process of numerical simulation in this paper, the selection of parameters is subject to field monitoring, and the calibration test is carried out by triaxial test, and the modeling parameters of the following soils are obtained. Triaxial test is a common method to determine the macroscopic parameters of soil. Unconsolidated and undrained test is carried out on a series of sand and clay with density gradient through the triaxial tester. The relationship between the internal friction Angle ϕ, cohesion force c, deformation modulus k and porosity is fitted, and the correlation between the shear strength and porosity of the soil is studied, which provides a basis for the parameter assignment of the subsequent shield model.

 Triaxial analysis test

3. In fact, DEM is not an appropriated numerical tool for modelling the process of tunnelling, given that it cannot mimicking simultaneous grouting. The soil loss rate is 6.15% based on the diameter of the tunnel excavation and the outer diameter of the segment. But adoption of this value will overestimate the disturbance tunnelling to the surrounding soil and pile foundation, as simultaneous grouting was not considered in 

---

## [Decision Letter · Decision Letter 1]

10 May 2024

PONE-D-24-03222R1Study on deformation of tunnel pile foundation based on discrete element method and finite difference method couplingPLOS ONE

Dear Dr. Dai,

Thank you for submitting your manuscript to PLOS ONE. After careful consideration, we feel that it has merit but does not fully meet PLOS ONE’s publication criteria as it currently stands. Therefore, we invite you to submit a revised version of the manuscript that addresses the points raised during the review process.

We look forward to receiving your revised manuscript.

Kind regards,

Mohammad Azadi

Academic Editor

PLOS ONE

Additional Editor Comments:

Unfortunately, the revision was not done properly and completely. As another last change, again, another revision must be addressed on the revised text based on two new reviewers' comments and the following issue,

1) The abstract is lengthy. It must be shortened.

2) The introduction is still lengthy.

3) All features must be mentioned on the image of Figure 2. Moreover, it is too small.

4) All used material properties need references, especially for the data in Tables 1 and 2. This comment was mentioned before and not addressed!

5) The verification of numerical results without any references or experiments has no meaning.

6) Based on the first reviewing, the reviewer asked the authors to provide some engineering background for the three-hole small distance tunnel construction near pile foundation. It must be added to the main text.

7) No proper answer was mentioned for the comment of the reviewer that the DEM is not an appropriated numerical tool for modelling the process of tunnelling. References must be used and descriptions must be added to the main text.

8) Again, the reviewer asked to mention the limitation of the study should be highlighted in the discussions and conclusions. Not done properly.

Reviewers' comments:

Reviewer's Responses to Questions

**Comments to the Author**

1. If the authors have adequately addressed your comments raised in a previous round of review and you feel that this manuscript is now acceptable for publication, you may indicate that here to bypass the “Comments to the Author” section, enter your conflict of interest statement in the “Confidential to Editor” section, and submit your "Accept" recommendation.

Reviewer #3: (No Response)

Reviewer #4: (No Response)

2. Is the manuscript technically sound, and do the data support the conclusions?

Reviewer #3: Yes

Reviewer #4: (No Response)

3. Has the statistical analysis been performed appropriately and rigorously? 

Reviewer #3: (No Response)

Reviewer #4: (No Response)

4. Have the authors made all data underlying the findings in their manuscript fully available?

Reviewer #3: Yes

Reviewer #4: (No Response)

5. Is the manuscript presented in an intelligible fashion and written in standard English?

Reviewer #3: Yes

Reviewer #4: (No Response)

6. Review Comments to the Author

Reviewer #3: 1. How does the DEM-FDM coupled numerical simulation method, utilizing software such as FLAC2D and PFC2D, contribute to understanding the deformation response mechanism of pile foundations during tunnel construction, specifically in the context of a three-hole small clear distance tunnel?

2. What were the key findings regarding the influence of different factors, such as the length of the pile foundation, the distance between the tunnel and pile foundation, and the small clear distance of the three-hole tunnel construction, on the deformation response of pile foundations? How did these factors affect vertical deformation, settlement displacement, and uplift peak values?

3. Can you elaborate on the observations made regarding the impact of varying pile lengths on the vertical deformation caused by tunnel construction and how settlement displacement and uplift peak values were distributed along the pile foundation in different scenarios?

4. How did the distance between the tunnel and pile foundation influence horizontal displacement differences between the pile top and bottom, and what implications did this have for the structural behavior and integrity of the pile foundation under tunnel construction loads?

5. What were the critical indications from the study regarding the distribution of axial stress on the pile body for short pile foundations and how did the incremental horizontal displacement at the pile head vary as the pile foundation approached the tunnel during the construction of the three-hole closely spaced tunnels?

6. How did the construction of three-hole closely spaced tunnels impact soil deformation in the horizontal direction, leading to the formation of distinct zones of horizontal displacement concentration? Which areas around the tunnel were identified as prominent locations for horizontal displacement concentration based on the study findings?

7. In what ways can the research outcomes presented in this study be applied to inform and improve the design and construction practices for shield tunneling projects that involve underpassing pile foundations? What considerations should engineers and practitioners take into account to mitigate deformation risks and ensure the safety and stability of pile-supported structures during tunnel construction activities?

Reviewer #4: The paper presents a study to report on the deformation patterns of nearby piles during construction of a three-holes, small-spaced subway tunnel. The paper is well written and the results are presented with proper justifications. The problem is often encountered in practice and therefore has the potential for publication on the condition that some clarifications are provided in the revised manuscript.

(1) Spaces should be added between figures and units in the manuscript.

(2) The manuscript uses “1.6D” as well as “4D” in lines 108 and 109, and an explanation of what parameter D stands for should have been provided at the time of its first appearance.

(3) In Figure 1 in the manuscript, the dimensional labeling on the upper right side of the model does not provide specific values.

(4) Velocity should be written as “m/s” or “m·s-1”. Some units in the manuscript are written in error, e.g., line 127, where “m·s-1” should not be used, and the authors are advised to verify this.

(5) In lines 213 and 214 of the manuscript, “9.946 meters” and “6.2 meters” should be written as “9.946 m” and “6.2 m”.

(6) The first letter of the vertical label in figure 6 should be capitalized, and “surface settlement” should be changed to “Surface settlement”.

(7) Lines 372 and 373 are too long as section headings for the article, and it is suggested that they be revised to read: “The influence of tunnel-pile spacing on the response of pile”. Lines 455 and 456 are too long as section headings for the article, and it is suggested that they be revised to read: “The effect of repeated excavation on pile”.

7. PLOS authors have the option to publish the peer review history of their article (what does this mean?). If published, this will include your full peer review and any attached files.

Reviewer #3: **Yes: **Prof. Dr. Mukhtiar Ali Soomro

Reviewer #4: **Yes: **Pin-Qiang Mo

---

## [Author Response · Author response to Decision Letter 1]

21 Jun 2024

Dear Editors and Reviewers,

We sincerely thank our editors and all reviewers for their valuable feedback, which has helped us to improve the quality of our manuscripts. Reviewer comments are listed in bold font and numbered for reference, and our responses are given in normal font. Changes/additions to the original text are shown in red font.

Editor:

1) The abstract is lengthy. It must be shortened.

We have condensed the abstract to 150 words. The revised abstract is as follows:

The deformation of pile caused by tunnel excavation will weaken the bearing capacity of the foundation. In order to investigate the deformation response of pile induced by the construction of three-hole small spacing tunnel, the DEM-FDM (discrete element method and finite difference method) coupling numerical simulation method were used to simulate the deformation process of pile during tunnel excavation. This paper probed into the deformation response of pile by three factors: the length of pile, the pile-tunnel spacing, and the three-hole tunnel construction. The results showed that as the pile-tunnel spacing decreases, the incremental horizontal displacement of the pile top became more significant when the three-hole tunnel was excavated. The excavation resulting in four zones of horizontal displacement concentration. The prominent locations were mainly concentrated on both sides of the tunnel and the ground directly above the tunnel. The research findings of this study can provide insights and references for the design and construction of shield tunneling underpassing piles.

2) The introduction is still lengthy.

We have further streamlined the language and removed redundant sentences from the manuscript.

3) All features must be mentioned on the image of Figure 2. Moreover, it is too small.

The main research approach employed in this study is numerical simulation analysis, focusing on the mechanism of repeated disturbances from closely spaced tunnels on adjacent piles. Figure 2 serves merely as a calibration experiment to validate the parameters used in the simulations, while the triaxial tests are conventional experimental methods. Therefore, the article does not extensively elaborate on their details. The original photograph for Figure 2 was taken earlier and the angle chosen was not ideal. We have opted to remove Figure 2, which does not compromise the integrity of the paper or subsequent analyses. We appreciate the editor's suggestions and efforts.

4) All used material properties need references, especially for the data in Tables 1 and Tables 2. This comment was mentioned before and not addressed!

We have supplemented the references according to the editor's suggestions to support the data presented in the tables. The data in Table 1 are derived from our own test results, with the cited references being previously published work by our team. The data in Table 2 are selected based on Chinese industry standards, and we have added the relevant standards as references.

5) The verification of numerical results without any references or experiments has no meaning.

This paper analyzes a typical working condition. Although three-line parallel small-spacing tunnels are rare, they do exist. Consequently, there are few prior studies on this topic, and the existing literature does not specifically address small-spacing tunnels. Therefore, this paper does not validate its results against existing literature. However, as subway construction expands, the frequency of such situations increases, making this study valuable for similar construction projects. In this paper, we primarily validate the model's reliability by comparing numerical simulation data with settlement data obtained from field monitoring (Figure 5). Similar studies in the existing literature also predominantly use monitoring data for model validation. The settlement curves obtained in this paper align well with the field-monitored settlement curves in terms of settlement area width and peak settlement, supporting further analysis in the paper. Additionally, for pile deformation patterns, the paper compares and explains the findings with those in references [27], [32], [30], [36] when analyzing Figures 8, 10, 11, 12, 15, and 19.

Fig.5 The results of numerical simulation verification

6) Based on the first reviewing, the reviewer asked the authors to provide some engineering background for the three-hole small distance tunnel construction near pile foundation. It must be added to the main text.

We have added the engineering background in the "Construction Plan Design" section of the paper. This includes basic information about the project, such as tunnel spacing, depth, diameter, soil type, and the actual construction sequence. Additionally, we have included a schematic diagram of the site tunnels. The revised content is as follows:

“Take an urban rail transit shield section in China as an example, the shield is a three-hole section, including the up line, the down line and the stop line. The designed start and end points for the uprunning line in this section are: SK20+226.052 to SK20+741.314, spanning a total length of 515.262 meters. Similarly, the designed start and end points for the downward line are: XK20+227.065 to XK20+741.330, covering a total length of 514.197 meters. The start and end points for the parking line in this section are: TK0+112.500 to TK0+627.514, with a total length of 515.014 meters. The vertical distance between the top and bottom lines is 22 meters. Therefore, this tunnel is a small clear distance tunnel. The maximum and minimum slope of the line are 4‰ and 2‰, respectively. The burial depth of the line ranges from 8.916 to 9.946 m, and the outer diameter of the segment is 6.2 m.

The line of the three-hole shield tunnel section travels along the existing road. Along the line are the city's main roads and important commercial blocks and residential areas. The underground pipeline along the line is dense and passes through the existing civil air defense tunnel. The actual scene diagram of the project case is shown in Fig. 4 (a).”

(a) Project case site layout

(b) Construction sequence of shield tunnel sections

Fig.4 Engineering design scheme

7) No proper answer was mentioned for the comment of the reviewer that the DEM is not an appropriated numerical tool for modelling the process of tunnelling. References must be used and descriptions must be added to the main text.

Numerical simulation of tunnel grouting has always been one of the challenging aspects of grouting technology research. However, some studies have successfully employed the Discrete Element Method (DEM) for the numerical analysis of filling grouting (see reference [32]). Additionally, DEM is very suitable for simulating discrete systems, such as sand and gravel soils, and can even simulate the cutting process during excavation. Therefore, it is widely used in tunnel construction-related research (see reference [30]; a search for "DEM" and "Tunnel" in Web of Science retrieves 382 related papers). The original intention of the reviewer was not to indicate that DEM is unsuitable for simulating tunnel excavation, but rather to emphasize that when using this method to analyze shield tunneling, it is important to appropriately address the issue of the tail gap. The introduction of the paper cites relevant literature to illustrate the prevalence of DEM in shield tunneling research, such as references [27], [30], [31], and [32]. Furthermore, the tunnels discussed in this paper are mainly excavated in sandy soils, and there is extensive research on using DEM to simulate the mechanical behavior of sandy soils. This paper specifically considers the advantages of DEM in simulating soil layers and the advantages of the finite element method(FEM) in simulating continuous media (concrete) to establish a numerical simulation model.

The reviewer pointed out that the article does not consider grouting, and therefore, using the theoretical ground loss rate might overestimate the impact of excavation on adjacent piles. The reviewer's comment is very insightful, and it is indeed an aspect that was not considered in this paper. Consequently, we have written a section in the conclusion that introduces considerations for further research based on this study, specifically mentioning the need to account for grouting when using DEM to study the impact of tunnel construction. On the other hand, since this paper primarily explores the influence of construction sequence, pile-tunnel distance, and pile length on the deformation of adjacent piles, the ground loss rate is kept constant for all conditions. Thus, the patterns discussed in the calculation results are not affected by the factors addressed in this paper. However, based on the reviewer's suggestion, we have added a supplement in the “DEM-FDM Coupling Model and Parameter Setting” section, explaining the basic assumptions and premises of our model. The specific content is as follows:

Introduction:

“Discrete element method (DEM), as an emerging numerical simulation method, have been applied in various fields[28–31]. In geotechnical engineering, it can simulate the interaction between particles or particle groups, and is suitable for simulating the behavior of granular materials such as soil. FDM, as an continuous medium method, can better describe the changes in continuous media and is more suitable for concrete continuity modeling. By coupling this two methods, the interaction between granularity and continuity can be considered comprehensively, so that the behavior of pile in the process of excavation can be simulated more accurately. The DEM-FDM coupling method was verified by Gholaminejad et al [26]. It is a robust method to simulate geotechnical problems. lee et al. [27] used DEM and FDM to conduct 3D numerical simulation of earth pressure balance (EPB) shield tunnel. The simulation results show that the numerical model based on DEM-FDM coupling can reasonably simulate tunnel shield tunneling under the condition of TBM operation, and has good robustness. Consequently, more and more research has based on DEM-FDM coupling method[28-30]. Qu et al. [31] established a computational model using the coupled DEM-FDM method and realized the simulation of the whole process of shield construction. By using the coupling method, better results can be obtained in the study of pile and tunnel[32,33].”

DEM-FDM coupling model and parameter setting:

“In order to investigate the soil deformation caused by subway tunnel excavation and the response of pile foundation deformation, this paper selects the finite difference program FLAC2D to build tunnel segments and adjacent pile foundation through grid modeling. PFC2D is used to deposit strata through particles. As the problems in computational efficiency of 3D computational analysis have not been solved, this paper currently discusses the horizontal deformation of piles only at the 2D level. A two-dimensional computational stratigraphic model with a width of 72m and a height of 26m was established, as shown in Fig. 1. The diameter of the tunnel excavation is 6.4m, the outer diameter of the segment is 6.2m, that is, the shield tail clearance is 10cm, the soil loss rate is 6.15%, and the thickness of the segment is 35cm according to the actual construction situation. In the process of accumulation modeling, in order to ensure the accuracy of particle part calculation. The number of particles should be reduced as much as possible to ensure that the computer can give full play to the calculation power. The particle size is 5cm, and the total number of particles before excavation is 84,117. In order to reflect the influence mechanism of buried depth of three-wire tunnel with small distance on pile foundation and stratum, the tunnel buried depth is set as 13.1m, the covering layer thickness at the top of the tunnel is 10m, the distance between the bottom of the segment and the bottom of the model is 10m (1.6D), and the distance between the outer side of the tunnel segments on both sides of the model is 25m (4D). It can ensure that the effect of model boundary does not affect the calculation result [34]. This chapter only discusses the geometric characteristics of pile foundation and the influence of the distance between pile foundation and tunnel, so the difference of different soil layers is ignored. The horizontal displacements of the piles obtained from the calculations may be large compared to the actual conditions because the effect of infill grouting on the soil loss rate was not considered. However, this does not affect the analysis of the effect of pile length and pile-tunnel spacing since the fill grouting was not considered for all the conditions.”

Discussions:

“This paper puts forward some meaningful conclusions, however, there are still some limitations. Firstly, the coupling models in this paper are all 2D methods. Although 2D methods can simulate deformation on a plane, in reality, the tunnelling process is a three-dimensional problem. In the future, researchers can use FLAC3D and PFC3D for coupling simulations. Secondly, synchronous grouting is not considered in the simulation process of this paper. Although the selection of tunnel excavation diameter and segment outside diameter in this paper is subject to the actual situation, there will be some deviation in the simulation results. In the future research, the numerical simulation of synchronous grouting process can be considered to improve the accuracy of simulation results. Finally, in the future, researchers can supplement the effects of soil properties, groundwater conditions, and tunnel construction methods to improve the depth and breadth of research.”

8) Again, the reviewer asked to mention the limitation of the study should be highlighted in the discussions and conclusions. Not done properly.

Thank you to the reviewers and the editor for your suggestions. In the previous round of review, the reviewers highlighted the article's limitations, specifically the lack of consideration for grouting and the influence of thickness direction, suggesting the need for further 3D computational analysis. We have already addressed the grouting issue in our previous response and included relevant discussions in the final paragraph of the conclusion. Regarding the 3D analysis, conducting large-scale DEM analyses in three dimensions requires substantial computational power and long computation times. Therefore, most engineering analyses using DEM in the literature primarily focus on 2D calculations. Although 3D computation is the ultimate goal in this field and we are actively exploring ways to enhance computational efficiency, it will only be reflected in our future research outcomes. Following the reviewers' suggestions, we have discussed these limitations in the final paragraph of the conclusion. Additionally, we have included relevant explanations in the "DEM-FDM Coupling Model and Parameter Setting" section to address readers' considerations when reviewing this paper. The specific content is as follows:

DEM-FDM coupling model and parameter setting:

“In order to investigate the soil deformation caused by subway tunnel excavation and the response of pile foundation deformation, this paper selects the finite difference program FLAC2D to build tunnel segments and adjacent pile foundation through grid modeling. PFC2D is used to deposit strata through particles. As the problems in computational efficiency of 3D computational analysis have not been solved, this paper currently discusses the horizontal deformation of piles only at the 2D level. A two-dimensional computational stratigraphic model with a width of 72m and a height of 26m was established, as shown in Fig. 1. The diameter of the tunnel excavation is 6.4m, the outer diameter of the segment is 6.2m, that is, the shield tail clearance is 10cm, the soil loss rate is 6.15%, and the thickness of the segment is 35cm according to the actual construction situation. In the process of accumulation modeling, in order to ensure the accuracy of particle part calculation. The number of particles should be reduced as much as possible to ensure that the computer can give full play to the calculation power. The particle size is 5cm, and the total numbe

---

## [Decision Letter · Decision Letter 2]

4 Jul 2024

Study on deformation of tunnel pile foundation based on discrete element method and finite difference method

PONE-D-24-03222R2

Dear Dr. Dai,

We’re pleased to inform you that your manuscript has been judged scientifically suitable for publication and will be formally accepted for publication once it meets all outstanding technical requirements.

Kind regards,

Mohammad Azadi

Academic Editor

PLOS ONE

Additional Editor Comments (optional):

Almost done!

Reviewers' comments:

Reviewer's Responses to Questions

**Comments to the Author**

1. If the authors have adequately addressed your comments raised in a previous round of review and you feel that this manuscript is now acceptable for publication, you may indicate that here to bypass the “Comments to the Author” section, enter your conflict of interest statement in the “Confidential to Editor” section, and submit your "Accept" recommendation.

Reviewer #3: All comments have been addressed

Reviewer #4: All comments have been addressed

2. Is the manuscript technically sound, and do the data support the conclusions?

Reviewer #3: Yes

Reviewer #4: Yes

3. Has the statistical analysis been performed appropriately and rigorously? 

Reviewer #3: N/A

Reviewer #4: Yes

4. Have the authors made all data underlying the findings in their manuscript fully available?

Reviewer #3: Yes

Reviewer #4: Yes

5. Is the manuscript presented in an intelligible fashion and written in standard English?

Reviewer #3: Yes

Reviewer #4: Yes

6. Review Comments to the Author

Reviewer #3: I am satisfied with the authors' responses on my comments. hence, I am inclined to accept the revised manuscript for publication in the Journal.

Reviewer #4: All my previous comments have been addressed adequately. Therefore, this paper is suggested to be accepted.

7. PLOS authors have the option to publish the peer review history of their article (what does this mean?). If published, this will include your full peer review and any attached files.

Reviewer #3: No

Reviewer #4: No

---

## [Editor Report · Acceptance letter]

10 Jul 2024

PONE-D-24-03222R2 

PLOS ONE

Dear Dr. Dai, 

I'm pleased to inform you that your manuscript has been deemed suitable for publication in PLOS ONE. Congratulations! Your manuscript is now being handed over to our production team.

Kind regards, 

on behalf of

Dr. Mohammad Azadi 

Academic Editor

PLOS ONE